# REEF: Representation Encoding Fingerprints for Large Language Models

**Jie Zhang[1,2]\*, Dongrui Liu[1]\*, Chen Qian[1,3], Linfeng Zhang[4], Yong Liu[3], Yu Qiao[1], Jing Shao[1]†**

[1] Shanghai Artificial Intelligence Laboratory  [2] University of Chinese Academy of Sciences
[3] Renmin University of China  [4] Shanghai Jiao Tong University

## ABSTRACT

Protecting the intellectual property of open-source Large Language Models (LLMs) is very important, because training LLMs costs extensive computational resources and data. Therefore, model owners and third parties need to identify whether a suspect model is a subsequent development of the victim model. To this end, we propose a training-free REEF to identify the relationship between the suspect and victim models from the perspective of LLMs' feature representations. Specifically, REEF computes and compares the centered kernel alignment similarity between the representations of a suspect model and a victim model on the same samples. This training-free REEF does not impair the model's general capabilities and is robust to sequential fine-tuning, pruning, model merging, and permutations. In this way, REEF provides a simple and effective way for third parties and models' owners to protect LLMs' intellectual property together. Our code is publicly accessible at `https://github.com/AI45Lab/REEF`.

## 1 INTRODUCTION

The training process of Large Language Models (LLMs) requires extensive computational resources and time. Therefore, open-source models are usually released with specific licenses (*e.g.*, Apache2.0, and LLaMA 2 Community License (Meta AI, 2023)) to protect their intellectual properties (IPs). Unfortunately, some developers claim to have trained their own LLMs but actually wrapped or fine-tuned based on other base LLMs (*e.g.*, Llama-2 and MiniCPM-V) (OpenBMB, 2023; 01-ai, 2023). It is urgent for model owners and third parties to identify *whether the suspect model is a subsequent development of the victim model that serves as the root origin (e.g., Codellama trained from Llama-2) or is developed from scratch (e.g., Mistral).*

The key is to extract unique features (*i.e.*, fingerprints) that can authenticate the victim model. Watermarking methods artificially inject triggers into the victim model to make it generate specific content for identification (Peng et al., 2023a; Xu et al., 2024). However, watermarks introduce extra training costs and impair the model's general capabilities (Russinovich & Salem, 2024), or even can be removed (Wang & Kerschbaum, 2019; Chen et al., 2023a). More crucially, these methods can not be applied to models that have already been open-released. An alternative is to extract intrinsic features of the victim model, avoiding additional training and the compromise of capabilities. Weight-based fingerprints are one of intrinsic features that allow calculating the similarity between a suspect model and a victim model's weights for identification (Zeng et al., 2023; Refael et al., 2024). However, these methods are fragile to major changes in weights, *e.g.*, weight permutations, pruning, and extensive fine-tuning (Fernandez et al., 2024; Xu et al., 2024). This necessitates extracting more robust intrinsic features as fingerprints to identify victim models and protect their IPs.

In this paper, we propose to solve this problem from the perspective of the *feature representations* of LLMs, beginning with the following visualization analysis. It is generally acknowledged that different models encode informative and intrinsic features based on their training data and model architecture, resulting in distinct feature representations across models (Mikolov et al., 2013; Bolukbasi et al., 2016; Karras et al., 2021; Chen et al., 2023b; Zou et al., 2023; Dang et al., 2024). Figure 1(a) illustrates that the representations of Llama are markedly distinct from those of Baichuan and Qwen, while largely comparable to its fine-tuned models (*i.e.*, Llama-chat and Chinese-llama).

---

\* Equal contribution    † Corresponding author

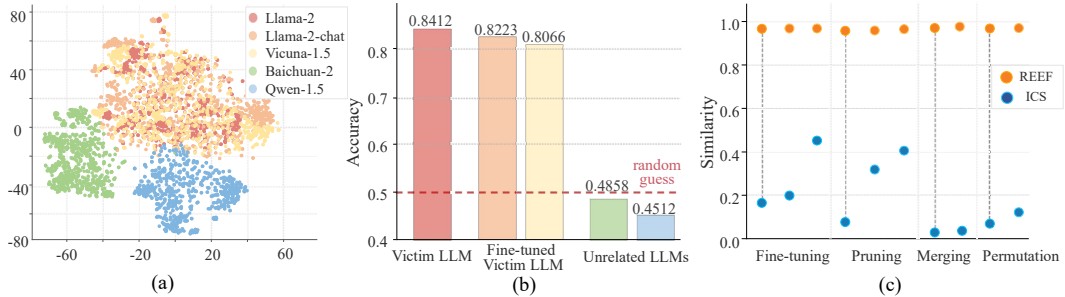

Figure 1: (a) t-SNE visualization of different LLMs' representations on the same samples. (b) Performance of classifiers trained on representations from the victim model evaluated on suspect models. (c) Robustness of REEF under variant LLMs that cause ICS (Zeng et al., 2023) ineffective.

Such findings inspire us to construct representation-based fingerprints. Specifically, we apply neural networks to extract fingerprints of a victim model from its representation space. Figure 1(b) shows that the classifier trained on representations of a victim model (*i.e.*, Llama) can be generalized to its variant models (*e.g.*, Llama-chat and Vicuna), but fail to other models (*e.g.*, Baichuan and Qwen). Although the effectiveness of representation-based fingerprints has been validated, such fingerprints still have limitations. On one hand, the input dimensions of neural networks are fixed, making them inapplicable to model pruning that alters the representation dimensions of the victim model (Frantar & Alistarh, 2023; Xia et al., 2023; 2024), which is prevalent in scenarios such as model compression for deployment on mobile devices. On the other hand, these fingerprints lack robustness against representation permutations, a challenging issue because developers may intentionally manipulate model representations to evade detection (Zeng et al., 2023; Refael et al., 2024).

To this end, we propose a simple and effective approach, namely REEF, which is robust against pruning and evading detection. Specifically, REEF is a representation-based fingerprinting method that compares the Centered Kernel Alignment (CKA) similarity (Kornblith et al., 2019) between the representations of the same samples from a suspect model and a victim model that serves as the root origin. Experimental results indicate that models derived from the victim model exhibit high similarity. Moreover, REEF is resilient to dimensional changes, and we theoretically prove that CKA is invariant to column-wise permutations and scaling transformations. Figure 1(c) demonstrates that REEF maintains its effectiveness even under extreme conditions that cause weight-based methods (Zeng et al., 2023) ineffective. These conditions include extensive fine-tuning (using data with up to 700B tokens (Azerbayev et al., 2023)), a high ratio pruning (up to 90% of parameters (Ma et al., 2023)), model merging (LLMs with different architectures (Wan et al., 2024a)), and permutations (parameter vector direction change through weight rearrangements (Fernandez et al., 2024)).

REEF utilizes the intrinsic feature from the perspective of representations to identify whether a suspect model is derived from a root victim model under the white-box scenario. This training-free REEF does not impair model's general capabilities and is robust to various subsequent developments compared to weight-based fingerprints and watermarks. Consequently, REEF is a promising method for protecting the IPs of model owners and provides an efficient and effective way for third parties to review models, combating unethical or illegal activities such as unauthorized use or reproduction.

## 2  RELATED WORK

Model fingerprinting protects IPs by allowing model owners and third parties to authenticate model ownership. There are two types of fingerprints for LLMs. One is injected fingerprints, which are artificially added during training or fine-tuning to facilitate model identification, such as watermarking methods (Peng et al., 2023a; Xu et al., 2024). The other is intrinsic fingerprints, which are inherent properties that naturally emerge from the models' training data and architectures, including model weights (*i.e.*, parameters) and feature representations, also known as embeddings or activations.

**Injected Fingerprint.**  Watermarking methods inject a backdoor trigger into a victim model, causing it to produce specific outputs when the trigger is present. This allows for identifying whether a suspect model derives from the victim model. Many approaches embed the watermarks through

backdoor attacks (Adi et al., 2018; Zhang et al., 2018; Li et al., 2019b), and digital signature technology and hash functions (Guo & Potkonjak, 2018; Li et al., 2019a; Zhu et al., 2020) are also used to design trigger words that contain the owner's identity information to protect the IPs of DNNs. For LLMs, the high computational and time costs of training pose an urgent need to protect their IPs. Researchers propose various methods to inject watermarks as fingerprints to identify the victim model (Li et al., 2023; Peng et al., 2023b; Kirchenbauer et al., 2023; Zhao et al., 2023; Russinovich & Salem, 2024; Xu et al., 2024), but such methods inevitably impair the model's overall performance.

**Intrinsic Fingerprint.** Such fingerprints use the inherent and native attributes of the victim model, without requiring additional tuning which could impair the model's general capabilities, and are more stable and can not be removed. Model weights are one of the intrinsic features that can be used to compute the similarity of parameters between a suspect model and a victim model for identification (Zeng et al., 2023; Refael et al., 2024). Semantic analysis methods conduct statistical analysis on the content generated by different models, exploiting the linguistic patterns and semantic preferences exhibited by various LLMs as their unique fingerprints (Iourovitski et al., 2024; Pasquini et al., 2024; McGovern et al., 2024). However, both methods suffer from insufficient robustness (Xu et al., 2024). The internal representations of LLMs are derived from the data, strategies, and frameworks used during the training process, and serve as intrinsic features for model identification (Sevastjanova et al., 2022). For example, the logits space can be leveraged to identify the victim model (Yang & Wu, 2024). However, this approach remains highly sensitive to parameter permutation, posing significant challenges for effective fingerprinting.

# 3 EXPLORING THE POTENTIAL OF FEATURE REPRESENTATIONS AS FINGERPRINTS

In this section, we propose to utilize *feature representations* as LLM fingerprints to identify whether a suspect model is a subsequent development of the victim model in a white-box scenario, based on the following two observations. (1) Feature representations of fine-tuned victim models are similar to feature representations of the original victim model, while the feature representations of unrelated models exhibit distinct distributions, as shown in Figure 1(a). (2) Some high-level semantic concepts are "linearly" encoded in the representation space of LLMs and can be easily classified, such as safety or unsafety and honest or dishonest (Zou et al., 2023; Slobodkin et al., 2023; Qian et al., 2024b; Lu et al., 2025). According to these two observations, we can train a binary classifier on the representations of the victim model and then apply it to various suspect models' representations, *i.e.,* LLMs derived from the victim model and unrelated LLMs. In this way, such a classifier may generalize to different fine-tuned victim models, because they have similar feature representations.

The binary classifier can employ various Deep Neural Network (DNN) architectures, such as a linear classifier, Multi-Layer Perceptron (MLP), Convolutional Neural Network (CNN), and Graph Convolutional Network (GCN). For training, we use the TruthfulQA dataset (Lin et al., 2022), concatenating each question with its truthful answer as positive samples and with its false answer as negative samples. The dataset is split into training and test sets with a 4:1 ratio. To evaluate the classifier's performance, we conduct experiments on LLMs of varying sizes. Specifically, we select Llama-2-7b and Llama-2-13b as the victim models, while derived models and unrelated LLMs serve as suspect models for comparison.

**Classifiers trained on representations of a victim model can effectively generalize to its variants but not to others.** Figure 2(a) shows that a classifier trained on the 18th layer representation of Llama-2-7b achieves approximately 80% classification accuracy when applied to its fine-tuned models (*e.g.*, Chinese-llama-2-7b). However, the accuracy drops to around 50% on unrelated models (*e.g.*, Mistral-0.1-7b), which is close to the level of random guessing. Classifiers trained on representations from other layers show the same results, as discussed in Appendix B. Additionally, similar findings are observed for Llama-2-13b (Figure 2(b)), indicating the scalability of the representation-based fingerprints. These experimental results indicate that representations can serve as fingerprints to protect the victim model's IP.

**Challenges to using the classifier for victim model identification:** (1) DNNs have fixed input dimensions and cannot be applied to models pruned from the victim model, *e.g.*, reducing representation dimensions. For example, the pruned models Sheared-llama-1.3b and Sheared-llama-2.7b

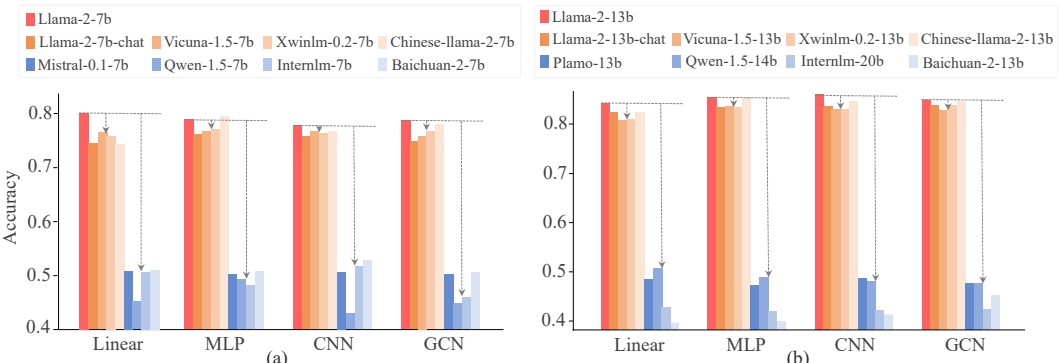

Figure 2: Accuracies of classifiers trained on representations from the victim model: (a) Llama-2-7b as the victim model, (b) Llama-2-13b as the victim model.

have dimensions of 2048 and 2560, respectively (Xia et al., 2024). However, the classifier trained on Llama-2-7b can only process inputs of 4096 dimensions. (2) DNNs are not robust to permutations of the input feature representations, such as when columns are permuted through coupled matrix multiplications, which malicious developers might use to evade detection (Fernandez et al., 2024).

# 4 ROBUST REPRESENTATION-BASED FINGERPRINTING WITH REEF

To address the challenges of classifiers in victim model identification, we propose REEF, an advanced representation-based fingerprinting approach for open-source LLMs that can adapt to suspect models with varying representation dimensions and is robust to representation permutations.

REEF identifies whether a suspect model is derived from a root victim model, given the representations of these two models on certain examples. Specifically, let $X \in \mathbb{R}^{m \times p_1}$ denote activations of the $l$-th layer from the suspect model on $m$ examples and $Y \in \mathbb{R}^{m \times p_2}$ denotes activations of the $l'$-th layers from the victim model on same $m$ examples, where $p_1$ is independent of $p_2$, meaning there is no limitation on dimensional consistency. Therefore, we need a similarity index $s(\cdot, \cdot)$ to measure representations' similarity between the suspect and victim models. In this way, a high $s(X, Y)$ score indicates that the suspect model is more likely derived from the victim model. In contrast, a low $s(X, Y)$ score means that the suspect model is less likely derived from the victim model.

**Centered Kernel Alignment.** CKA (Kornblith et al., 2019) is a similarity index based on Hilbert-Schmidt Independence Criterion (HSIC) (Gretton et al., 2005), which measures the independence between two sets of random variables. The CKA similarity between $X$ and $Y$ can be computed as follows

$$\text{CKA}(X, Y) = \frac{\text{HSIC}(X, Y)}{\sqrt{\text{HSIC}(X, X) \cdot \text{HSIC}(Y, Y)}}, \tag{1}$$

where $\text{HSIC}(X, Y) = \frac{1}{(m-1)^2}\text{tr}(K_X H K_Y H)$. Specifically, $H = I - \frac{1}{m}\mathbf{1}\mathbf{1}^{\text{T}}$ is a centering matrix. $K_X$ and $K_Y$ are Gram matrices that measure the similarity of a pair of examples based on kernel function $k$, i.e., $(K_X)_{ij} = k(X_i, X_j)$ and $(K_Y)_{ij} = k(Y_i, Y_j)$. $X_i$ and $X_j$ denote the $i$-th and $j$-th row of $X$, respectively.

**Kernel Selection.** In this study, we consider a linear kernel and a Radial Basis Function (RBF) kernel. In the linear kernel case, Gram matrix $K_X = XX^{\top}$. In the RBF kernel case, $k(X_i, X_j) = \exp(-||X_i - X_j||_2^2/(2\sigma^2))$. Empirically, we discover that linear and RBF kernels obtain similar experimental results. Please see Section 5.1 for more discussions. Unless otherwise specified, we adopt linear CKA due to its high efficiency.

**Theorem 1** *(Proof in Appendix A) Given two matrices $X \in \mathbb{R}^{m \times p_1}$ and $Y \in \mathbb{R}^{m \times p_2}$, the CKA similarity score between $X$ and $Y$ is invariant under any permutation of the columns and column-wise scaling transformation. Formally, we have:*

$$CKA(X, Y) = CKA(XP_1, YP_2) = CKA(c_1 X, c_2 Y) \tag{2}$$

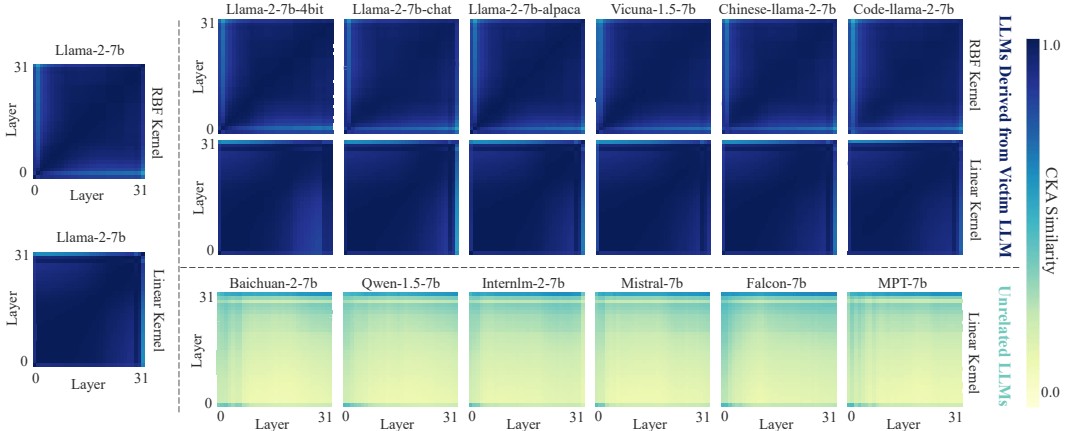

Figure 3: Heatmaps depicting the CKA similarity between the representations of the victim LLM (Llama-2-7B) and those of various suspect LLMs on the same samples.

where $P_1 \in \mathbb{R}^{p_1 \times p_1}$ and $P_2 \in \mathbb{R}^{p_2 \times p_2}$ denote permutation matrices. $c_1 \in \mathbb{R}^+$ and $c_2 \in \mathbb{R}^+$ are two positive scalars.

Theorem 1 indicates that the CKA similarity score is theoretically invariant and robust to any column-wise permutations and scaling transformations. Kornblith et al. (2019) have shown that CKA is able to the correspondence between representations of different dimensions. Therefore, REEF is highly robust to various subsequent developments of the victim model in practical scenarios, including model pruning and representation permutation, ensuring accurate identification of the victim model through representation-based fingerprints to protect its IP.

## 5 EXPERIMENTS

In this section, we provide a comprehensive evaluation of REEF. Section 5.1 evaluates REEF's effectiveness in distinguishing LLMs derived from the root victim model from unrelated models. Following this, Section 5.2 assesses REEF's robustness to subsequent developments of the victim model, such as fine-tuning, pruning, merging, and permutations. Section 5.3 presents ablation studies on REEF across varying sample numbers and datasets. Finally, Section 5.4 discusses REEF's sensitivity to training data and its capacity for adversarial evasion.

### 5.1 EFFECTIVENESS VERIFICATION

In this subsection, we demonstrate that REEF can effectively model the fingerprint from the representation. The CKA similarity between the victim model's representations and those of its derived models, as well as unrelated models, shows significant differences. This makes REEF a reliable fingerprinting method for protecting the victim model's IP.

**Settings.** For the LLMs, we select Llama-2-7b as the victim model and choose a range of suspect models, including quantization and fine-tuned variants of Llama-2-7b (*e.g.*, Llama-2-7b-chat, Code-llama-7b, and Llama-2-7b-4bit) as well as unrelated models (*e.g.*, Qwen-1.5-7b, Baichuan-2-7b, and Mistral-7b). We use both a linear kernel and an RBF kernel to compute the layer-wise and inter-layer CKA similarity of representations between the victim and suspect models on 200 samples from the TruthfulQA dataset (Lin et al., 2022). The resulting heatmap is shown in Figure 3.

**REEF can accurately distinguish between models derived from the victim model and unrelated models.** As shown in Figure 3, for LLMs derived from the victim model, the CKA similarity with the victim model is high (higher than 0.8), whereas unrelated LLMs show low similarity (lower than 0.5). This is reflected in the marked color contrast between the first two rows and the third row. To quantify results, the average similarity of LLMs derived from the victim model is 0.9585, which is higher than that of unrelated LLMs, whose average similarity is 0.2361. Additionally, for LLMs derived from the victim model, the similarity is notably high along the diagonal of the heatmaps,

which represents the similarity between corresponding layers of the victim and suspect models, with an average of 0.9930. Furthermore, the inter-layer similarity is also significant, reaching 0.9707.

**Linear and RBF kernels yield similar results in identifying whether a suspect model is derived from the victim model.** As shown in the first two rows of Figure 3, the CKA similarity between the victim model and the LLMs derived from it, calculated using both the linear and RBF kernels, exceeded 0.95. This demonstrates that both kernels are suitable for fingerprinting in REEF. We adopt the linear CKA due to its higher computational efficiency.

**CKA from a single layer is sufficient for fingerprint identification.** The similarities between representations on a specific layer of the victim model and those of the derived and unrelated models differ significantly (*e.g.*, 0.9973 and 0.2223 for layer 18, respectively). Consequently, we focus on reporting the similarity at layer 18 in subsequent experiments, due to its informativeness and efficiency. The complete heatmap results are provided in Appendix C.

## 5.2 ROBUSTNESS VERIFICATION

In this subsection, we apply REEF to suspect models that are developed from a victim model through fine-tuning, pruning, merging, permutations, and scaling transformations (Appendix D provides REEF's application across more different LLM families, including Qwen and Mistral). These techniques can introduce significant changes to the model's structure or parameters, making it challenging for existing methods to identify the victim model. However, REEF remains effective in these scenarios, demonstrating its robustness.

### 5.2.1 BASELINE METHODS

**Weight-based Fingerprinting Methods.** Following Zeng et al. (2023), we use model weight similarity methods, including PCS and ICS, to identify whether a suspect model is derived from a victim model. Specifically, PCS flattens all weight matrices and biases of an LLM into vectors and directly compares the cosine similarity between these vectors for the two models. ICS constructs invariant terms from the weights of the last two layers and calculates the cosine similarity between these invariant terms for the two models. A high cosine similarity indicates that the suspect model is derived from the victim model, and vice versa.

**Representation-based Fingerprinting Methods.** Yang & Wu (2024), referring to the Logits method, implements LLM fingerprinting by analyzing unique attributes of each LLM's logits output. This method evaluates the similarity between the output spaces of the victim and suspect models. A high similarity suggests that the suspect model is derived from the victim model. We conduct experiments on the TruthfulQA dataset to extract logit output for the suspect models.

### 5.2.2 FINE-TUNING

Xu et al. (2024) point out that weight-based fingerprints are not reliable when models undergo extensive fine-tuning with larger deviations in parameters. Given this challenge, we seek to assess the robustness of REEF under such demanding scenarios.

**Settings.** We use Llama-2-7b as the victim model and select a diverse set of its fine-tuned models as suspect models, with fine-tuning (FT) data volumes ranging from 5 million to 700 billion tokens. The suspect models include Llama-2-finance-7b, Vicuna-1.5-7b, Wizardmath-7b, Chinese-llama-2-7b, Code-llama-7b, and Llemma-7b, with each model's fine-tuning data volume being 5M, 370M, 1.8B, 13B, 500B, and 700B tokens, respectively (Chiang et al., 2023; Luo et al., 2023; Cui et al., 2023; Roziere et al., 2023; Azerbayev et al., 2023).

**REEF is robustness to extensive fine-tuning.** As shown in Table 1, even for models fine-tuned on datasets with up to 700B tokens (*i.e.*, Llemma-7B), REEF still achieves a high similarity of 0.9962. In contrast, PCS becomes ineffective as early as fine-tuning with 1.8B tokens (*i.e.*, Wizardmath-7b). ICS performance significantly degrades with increasing fine-tuning data volume, with 13B tokens (*i.e.*, Chinese-llama-2-7b) and 500B tokens (*i.e.*, Code-llama-7B) yielding similarity of 0.4996 and 0.2550, respectively. Although the Logits method shows relatively less degradation, it still exhibits sensitivity to the volume of fine-tuning data. Notably, Logits method is particularly sensitive to changes in the vocabulary, *e.g.*, Chinese-llama-2-7b has expanded its vocabulary during fine-tuning,

Table 1: Similarity of various LLM fingerprinting methods applied to suspect models developed through fine-tuning, pruning, merging, permutations, and scaling transformations. In this table, indicate similarity greater than 0.8, indicate similarity between 0.5 and 0.8, and indicate similarity less than 0.5.

| | Model Fine-tuning | | | | | |
|---|---|---|---|---|---|---|
| | Llama-2-finance-7b (5M Tokens) | Vicuna-1.5-7b (370M Tokens) | Wizardmath-7b (1.8B Tokens) | Chinesellama-2-7b (13B Tokens) | Codellama-7b (500B Tokens) | Llemma-7b (700B Tokens) |
| PCS | 0.9979 | 0.9985 | 0.0250 | 0.0127 | 0.0105 | 0.0098 |
| ICS | 0.9952 | 0.9949 | 0.9994 | 0.4996 | 0.2550 | 0.2257 |
| Logits | 0.9999 | 0.9999 | 0.9999 | 0.7033 | 0.7833 | 0.6367 |
| REEF | 0.9950 | 0.9985 | 0.9974 | 0.9979 | 0.9947 | 0.9962 |
| | Structured Pruning | | | | | |
| | Sheared-llama-1.3b-pruned | Sheared-llama-1.3b | Sheared-llama-1.3b-sharegpt | Sheared-llama-2.7b-pruned | Sheared-llama-2.7b | Sheared-llama-2.7b-sharegpt |
| PCS | 0.0000 | 0.0000 | 0.0000 | 0.0000 | 0.0000 | 0.0000 |
| ICS | 0.4927 | 0.3512 | 0.3510 | 0.6055 | 0.4580 | 0.4548 |
| Logits | 0.9967 | 0.9999 | 0.9999 | 0.9967 | 0.9999 | 0.9999 |
| REEF | 0.9368 | 0.9676 | 0.9710 | 0.9278 | 0.9701 | 0.9991 |
| | Unstructured Pruning | | | Distribution Merging (Fusechat-7b) | | |
| | Sparse-llama-2-7b | Wanda-llama-2-7b | GBLM-llama-2-7b | Internlm2-chat-20b | Mixtral-8x7b-instruct | Qwen-1.5-chat-72b |
| PCS | 0.9560 | 0.9620 | 0.9616 | 0.0000 | 0.0000 | 0.0000 |
| ICS | 0.9468 | 0.9468 | 0.9478 | 0.1772 | 0.0105 | 0.0635 |
| Logits | 0.9999 | 0.9999 | 0.9999 | 0.0000 | 0.0000 | 0.0000 |
| REEF | 0.9985 | 0.9986 | 0.9991 | 0.9278 | 0.9701 | 0.9991 |
| | Weight Merging (Evollm-jp-7b) | | | Distribution Merging(Fusellm-7b) | | |
| | Shisa-gamma-7b-v1 | Wizardmath-7b-1.1 | Abel-7b-002 | Llama-2-7b | Openllama-2-7b | Mpt-7b |
| PCS | 0.9992 | 0.9990 | 0.9989 | 0.9997 | 0.0194 | 0.0000 |
| ICS | 0.9992 | 0.9988 | 0.9988 | 0.1043 | 0.2478 | 0.1014 |
| Logits | 0.9933 | 0.9999 | 0.9999 | 0.9999 | 0.0100 | 0.0000 |
| REEF | 0.9635 | 0.9526 | 0.9374 | 0.9996 | 0.6713 | 0.6200 |
| | Permutation | | | Scaling Transformation | | |
| | Llama-2-7b | Mistral-7b | Qwen-1.5-7b | Llama-2-7b | Mistral-7b | Qwen-1.5-7b |
| PCS | 0.0000 | 0.0000 | 0.0000 | 0.9999 | 0.9989 | 0.9999 |
| ICS | 0.1918 | 0.9847 | 0.9912 | 0.9999 | 0.9999 | 0.9998 |
| Logits | 0.0000 | 0.0000 | 0.0000 | 0.9999 | 0.9999 | 0.9999 |
| REEF | 1.0000 | 1.0000 | 1.0000 | 1.0000 | 1.0000 | 1.0000 |

yielding a lower similarity than Code-llama-7b (0.7033 vs 0.7833), despite being fine-tuned on a smaller dataset (13B vs 500B tokens).

**Discussion about how much fine-tuning data could make REEF ineffective.** Despite fine-tuning Llama-2-7b to Llemma-7b with 700B tokens (Azerbayev et al., 2023), the fine-tuning data is one-third of Llama-2-7b's 2T token pre-training data, yet REEF remains effective. We question whether REEF would remain effective with continued increases in fine-tuning data. Before delving into this discussion, two statements are listed: (1) To the best of our know, Llemma-7b is the most extensively fine-tuned Llama-2-7b model, nearly 700B tokens for fine-tuning, and REEF has shown robustness in this context; (2) Code-llama-7b (Roziere et al., 2023) reports that fine-tuning with 500B tokens requires 4.4T of disk size and 25,000 GPU hours, fine-tuning on this scale is costly. Such a considerable cost limits further extensive fine-tuning. REEF appears effective in current fine-tuning scenarios.

### 5.2.3 MODEL PRUNING

Pruning is widely used in model compression for edge deployment, *e.g.*, serving for mobile devices and autonomous driving (Vadera & Ameen, 2021; Wang et al., 2024; Lin et al., 2024). However, pruning could significantly alter both the structural integrity and representation dimensions of models (Ma et al., 2023; Frantar & Alistarh, 2023; Zhu et al., 2023), posing challenges for fingerprint identification. To this end, we test REEF on various pruned models of the victim model Llama-2-7b.

**Settings.** We use Llama-2-7b as the victim model and various pruned models of it as suspect models. First, we select several pruned models using different pruning strategies, including structured pruning (*e.g.*Sheared-llama (Xia et al., 2024)), and unstructured pruning (*e.g.*, SparseGPT (Frantar & Alistarh, 2023), GBLM-Pruner (Das et al., 2023), and Wanda (Sun et al., 2023)). These methods prune the models at specific ratios, followed by post-training (*e.g.*, continued pre-training or

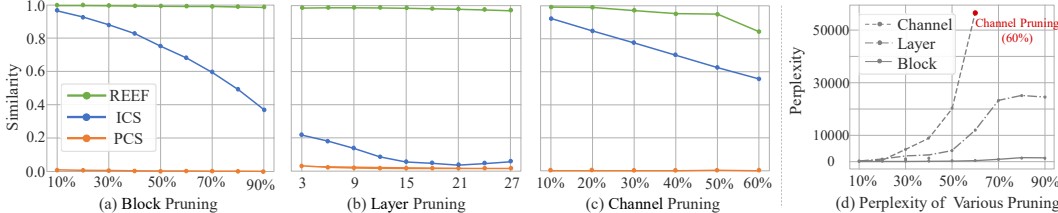

Figure 4: (a)-(c) Similarity between pruned models and the victim model across three pruning strategies at various pruning ratios. (d) Perplexity of the three pruning strategies.

instruction-tuning) to ensure the pruned models maintain their capabilities. Second, we apply LLM-Pruner (Ma et al., 2023) to prune Llama-2-7b into smaller suspect models at arbitrary pruning ratios, without post-training. For example, we apply block pruning to reduce Llama-2-7b's parameters by 10% to as much as 90%, and layer pruning to reduce the number of layers by 3 to as much as 27.

**REEF is robust to various pruning strategies.** As shown in Table 1, for structured pruned models, REEF consistently achieves accurate fingerprint identification across all Sheared-llama models, with similarities exceeding 0.9278. In contrast, PCS fails in this scenario, consistently yielding a similarity score of zero. ICS does not perform well, *e.g.*, the similarity for the 1.3B pruned model drops to 0.3512. The Logits method, which relies on the output space, remains unaffected unless the pruning alters the logits themselves. For unstructured pruned models, all methods are capable of identifying the victim model, with all similarities exceeding 0.94. In summary, REEF and the Logits method remain robust across all pruned models.

**REEF is robustness to pruning ratio, even up to 90%.** Figure 4 shows that REEF remains effective even with significant pruning, including block pruning of up to 90% of parameters, layer pruning of up to 27 layers, and channel pruning of up to 60%. Figure 4(d) illustrates that perplexities are particularly high in these scenarios, especially with 60% channel pruning. As noted by Ma et al. (2023), channel pruning affects all layers, but the first and last layers are critical for maintaining model integrity, thus pruning is limited to 60%. In contrast, PCS fails in all pruning scenarios, and ICS's effectiveness diminishes as the pruning ratio increases, ultimately failing under layer pruning. These findings highlight REEF as the most robust and reliable method for fingerprint identification across various pruning ratios.

### 5.2.4 MODEL MERGING

Model merging is an effective technique that merges multiple separate models with different capabilities to build a universal model without needing access to the original training data or expensive computation (Yang et al., 2024). Differing from other sections, the merged model is derived from several victim models, which pose a challenge in identifying all of them. In this subsection, we study two types of model merging: weight-based and distribution-based.

**Settings.** For weight merging, we select Evollm-jp-7b (Akiba et al., 2024) as the suspect model, which merges three victim models with the same architecture (*i.e.*, Shisa-gamma-7b-v1, Wizardmath-7b-1.1, and Abel-7b-002) by weighted parameters. For distribution merging, we choose Fusellm-7b (Wan et al., 2024a) and Fusechat (Wan et al., 2024b) as suspect models, respectively. Fusellm-7b merges three victim LLMs with distinct architectures but with same scale: Llama-2-7b, Openllama-2-7b, and Mpt-7b. Fusechat merges several chat LLMs of varied architectures and scales, we investigate Internlm2-chat-20b, Mixtral-8x7b-instruct, and Qwen-1.5-chat-72b as suspect models.

**REEF is robust across both weight and distribution merging scenarios.** For weight merging, REEF consistently achieves high accuracy in identifying the origins of merged models, with similarities ranging from 0.9526 to 0.9996, as shown in Table 1. ICS, PCS, and the Logits method also perform well in this scenario. For distribution merging at the same scales (*i.e.*, Fusellm-7b), REEF continues to perform well, accurately identifying the victim model Llama-2-7b with a similarity of 0.9996. Additionally, it remains effective for Openllama-2-7b and Mpt-7b, with similarities of 0.6713 and 0.62, respectively. However, ICS struggles significantly in this scenario, with all three original victim models achieving low similarities. Although PCS and the Logits method can iden-

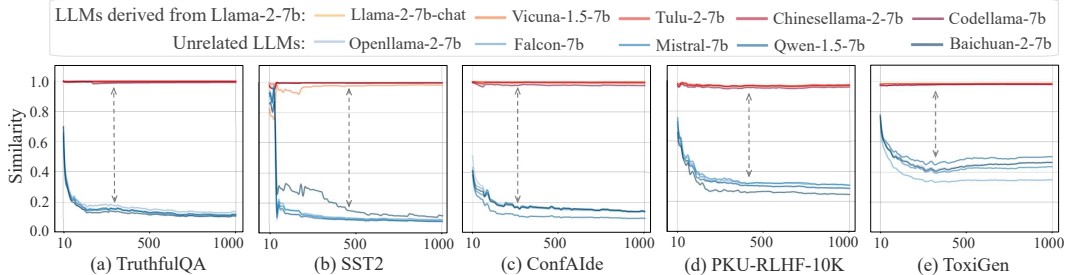

Figure 5: Illustration of the CKA similarity between the representations of the victim LLM (Llama-2-7B) and various suspect LLMs across different datasets as sample number increases.

tify Llama-2-7b, their performance drops sharply for Openllama-2-7b and Mpt-7b, with similarities of nearly 0. For distribution merging at the different scales (*i.e.*, Fusechat-7b), REEF is the only method that continues to work for identifying victim models, while the other methods fail, demonstrating its consistent reliability in this scenario. Based on these findings, REEF is robust across various merging strategies and can identify all victim models for the merged model.

### 5.2.5 PERMUTATION AND SCALING TRANSFORMATION

There are approaches that could camouflage the model without changing its architecture or affecting its output (Zeng et al., 2023). Malicious developers may modify the model by employing dimension permutation or coupled matrix multiplications to evade some fingerprint detection methods (Fernandez et al., 2024). This section aims to experiment with the robustness of various fingerprinting methods in addressing this type of evasion.

**Settings.** We select Llama-2-7b, Mistral-7b, and Qwen-1.5-7b as victim models, applying column-wise permutations or scaling transformations (with a scaling factor of 0.8) to both their weight matrices and feature representations. These operations simulate evasion techniques that malicious developers might use, enabling us to compare the similarities of the weights and representations before and after the operations.

**REEF is invariant and robust to any column-wise permutations and scaling transformations, as proved by the Theorem 1.** As shown in Table 1, the CKA similarity computed by REEF remains consistently at 1 before and after the permutation or scaling transformations, indicating that REEF is invariant to these operations and robust against evasion techniques. However, other methods such as ICS, PCS, and Logits, while robust to scaling transformations, exhibit a significant drop in similarity under permutation, with values nearly dropping to 0. These results further reinforce that REEF is a highly reliable fingerprinting method in practical applications against malicious developers.

### 5.3 ABLATION STUDY

*Number of Samples* To evaluate the impact of sample number on the performance of REEF, we conduct an ablation study using samples from TruthfulQA, ranging from 10 to 1000 in intervals of 10. We use Llama-2-7b as the victim model and select 10 suspect models, consisting of 5 LLMs derived from Llama-2-7b and 5 unrelated LLMs. We then calculate the CKA similarity between the sample representations of each suspect model and those of Llama-2-7b at different sample numbers. Figure 5(a) illustrates the similarities for various models as the number of samples increases.

**REEF is highly efficient regarding the number of samples required for robust model fingerprinting.** Figure 5(a) shows that the similarities for most models stabilize after 200-300 samples, suggesting that REEF can achieve reliable fingerprint identification with a smaller sample number. Notably, LLMs derived from Llama-2-7b (*e.g.*, Chinese-lama-2-7b and Code-llama-7b) consistently maintain high similarities close to 1.0 across all sample numbers. This indicates that these models potentially share the same representation space as the victim model, verifying that representation is an intrinsic feature for fingerprinting. In contrast, unrelated LLMs (*e.g.*, Qwen-7b-v1.5 and Mistral-7b) exhibit lower similarities that gradually decrease and stabilize at levels below 0.2 as the number of samples increases. This suggests that these models are more distinct and require a larger num-

Figure 6: Heatmaps depicting the CKA similarity between the representations of (a) Llama-2-7b itself, and (b) paired LLMs with the same architecture but different pre-training data.

ber of samples for accurate fingerprinting. Overall, few samples from TruthfulQA are effective for REEF in identifying LLMs derived from the victim model compared to unrelated LLMs.

***Different Datasets*** To assess the effectiveness of REEF across various data types, we also conduct experiments using SST2 (Socher et al., 2013), ConfAIde (Mireshghallah et al., 2023), PKU-SafeRLHF (Ji et al., 2024), and ToxiGen (Hartvigsen et al., 2022). Following the same settings described in the previous section, we plot the similarities between the victim model and various suspect models for different datasets as the number of samples increases, as shown in Figure 5(b)-(e).

**REEF is effective across various datasets.** Figure 5(b)-(e) show that the similarity between the victim model and its derived LLMs is significantly higher than the similarity with unrelated LLMs across different datasets. This clear distinction demonstrates that REEF can effectively identify whether a suspect model is derived from the victim model. Furthermore, the gap in the similarity between derived LLMs and unrelated LLMs varies by dataset, *e.g.*, the gap is approximately 0.8 on TruthfulQA and about 0.5 on ToxiGen. A larger gap indicates a stronger identification capability. Our findings suggest that while REEF is effective across diverse datasets, TruthfulQA emerges as the optimal choice for model fingerprinting, as it exhibits the most substantial differentiation in similarity between LLMs derived from the victim model and unrelated LLMs.

## 5.4 FURTHER DISCUSSION

**REEF can distinguish between models with the same architecture but different pre-training data.** Openllama-7b (Geng & Liu, 2023) and Amber (Liu et al., 2023) are open-source LLMs that use the same Llama architecture but are pre-trained on distinct pre-training corpus. In contrast to Figure 6(a), which shows that the layer-wise CKA similarity between Llama-2-7b itself is almost 1, Figure 6(b) clearly demonstrates that REEF effectively identifies the differences between Llama-2-7b and both Openllama-7b and Amber. Similar results are observed across different LLM generations, such as Llama-2-7b versus Llama-3-8b, and Internlm-7b versus Internlm2-7b. Each of these models reflects variations in pre-training data and strategies, which REEF accurately identifies.

**Malicious developers fail to fine-tune models with a customized loss function to evade detection by the REEF.** We assume these developers are aware of the REEF approach and attempt to design customized loss functions during fine-tuning to bypass detection. Since REEF relies on the observation that developed LLMs share similar representational spaces with the victim model. The developer may use the customized loss function to widen the gap between the two representations. Experimental results in Appendix E indicate that such fine-tuning seriously damage the model's general capabilities and renders the fine-tuned models unusable. This is because the capabilities of LLMs stem from their representational distribution, and such intentional fine-tuning inevitably leads to the model losing its language modeling ability. Therefore, malicious developers are unable to evade REEF detection through this method.

## 6 CONCLUSION

This paper proposes REEF, a robust representation-based fingerprinting method for LLMs in a white-box scenario, which effectively identifies models derived from a victim model that serves as the root origin. REEF does not impair LLMS's general capability and remains resilient against various subsequent developments, including pruning, fine-tuning, merging, and permutations. Therefore, REEF is highly suitable for protecting model IPs for both third parties and model owners, as a reliable solution for safeguarding models from unauthorized use or reproduction.

REPRODUCIBILITY STATEMENT

To ensure the reproducibility of this study, we have uploaded the source code as part of the supplementary material. Furthermore, the code and datasets will be made available on GitHub after the completion of the double-blind review process, enabling others to replicate our study.

ACKNOWLEDGMENTS

This work is supported by the Shanghai Artificial Intelligence Laboratory (No. JF-P23KK00072-1-DF). We also acknowledge the support of National Natural Science Foundation of China (No.62476277), CCF-ALIMAMA TECH Kangaroo Fund (No.CCF-ALIMAMA OF 2024008), and Huawei-Renmin University joint program on Information Retrieval. We also acknowledge the support provided by the fund for building worldclass universities (disciplines) of Renmin University of China and by the funds from Beijing Key Laboratory of Big Data Management and Analysis Methods, Gaoling School of Artificial Intelligence, Renmin University of China, from Engineering Research Center of Next-Generation Intelligent Search and Recommendation, Ministry of Education, from Intelligent Social Governance Interdisciplinary Platform, Major Innovation & Planning Interdisciplinary Platform for the "DoubleFirst Class" Initiative, Renmin University of China, from Public Policy and Decision-making Research Lab of Renmin University of China, and from Public Computing Cloud, Renmin University of China.

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

## A    PROOF FOR THEOREM 1

**Theorem 1** *Given two matrices $X \in \mathbb{R}^{m \times p_1}$ and $Y \in \mathbb{R}^{m \times p_2}$, the CKA similarity score between $X$ and $Y$ is invariant under any permutation of the columns and column-wise scaling transformation. Formally, we have:*

$$CKA(X, Y) = CKA(XP_1, YP_2) = CKA(c_1 X, c_2 Y) \tag{1}$$

*where $P_1 \in \mathbb{R}^{p_1 \times p_1}$ and $P_2 \in \mathbb{R}^{p_2 \times p_2}$ denote permutation matrices. $c_1 \in \mathbb{R}^+$ and $c_2 \in \mathbb{R}^+$ are two positive scalars.*

*Proof.*

### A.1    CASE 1: PERMUTATION INVARIANCE

**For Linear CKA**, the Gram matrices of $X$ and $Y$ are $K_X = XX^\top$ and $K_Y = YY^\top$, respectively. In this way, we have

$$K_{XP_1} = (XP_1)(XP_1)^\top = X \underbrace{P_1 P_1^\top}_{=I} X^\top = XX^\top = K_X. \tag{2}$$

Since $P_1$ is an orthogonal permutation matrix, thus $P_1 P_1^\top = I$.
Similarly, we have

$$K_{YP_2} = (YP_2)(YP_2)^\top = Y \underbrace{P_2 P_2^\top}_{=I} Y^\top = YY^\top = K_Y. \tag{3}$$

According to (Gretton et al., 2005),

$$
\begin{aligned}
\mathrm{HSIC}(X, Y) &= \frac{1}{(m-1)^2} \mathrm{tr}(K_X H K_Y H) \\
&= \underbrace{\frac{1}{(m-1)^2} \mathrm{tr}(K_{XP_1} H K_Y H)}_{\mathrm{HSIC}(XP_1, Y)} \\
&= \underbrace{\frac{1}{(m-1)^2} \mathrm{tr}(K_X H K_{YP_2} H)}_{\mathrm{HSIC}(X, YP_2)} \\
&= \underbrace{\frac{1}{(m-1)^2} \mathrm{tr}(K_{XP1} H K_{YP_2} H)}_{\mathrm{HSIC}(XP_1, YP_2)} \tag{4}
\end{aligned}
$$

Thus, we have

$$\mathrm{HSIC}(X, Y) = \mathrm{HSIC}(XP_1, Y) = \mathrm{HSIC}(X, YP_2) = \mathrm{HSIC}(XP_1, YP_2). \tag{5}$$

Taking Eq.5 into Eq. 1, we have

$$
\begin{aligned}
\text{CKA}(X, Y) &= \frac{\text{HSIC}(X, Y)}{\sqrt{\text{HSIC}(X, X) \cdot \text{HSIC}(Y, Y)}} \\
&= \underbrace{\frac{\text{HSIC}(XP_1, Y)}{\sqrt{\text{HSIC}(XP_1, XP_1) \cdot \text{HSIC}(Y, Y)}}}_{\text{CKA}(XP_1, Y)} \\
&= \underbrace{\frac{\text{HSIC}(X, YP_2)}{\sqrt{\text{HSIC}(X, X) \cdot \text{HSIC}(YP_2, YP_2)}}}_{\text{CKA}(X, YP_2)} \\
&= \underbrace{\frac{\text{HSIC}(XP_1, YP_2)}{\sqrt{\text{HSIC}(XP_1, XP_1) \cdot \text{HSIC}(YP_2, YP_2)}}}_{\text{CKA}(XP_1, YP_2)}
\end{aligned}
\tag{6}
$$

Finally, we obtain

$$
\text{CKA}(X, Y) = \text{CKA}(XP_1, Y) = \text{CKA}(X, YP_2) = \text{CKA}(XP_1, YP_2)
\tag{7}
$$

**For RBF CKA**, the RBF kernel function is

$$
\begin{aligned}
k(X_i, X_j) &= \exp\left(-\frac{\|X_i - X_j\|_2^2}{2\sigma^2}\right) \\
&= \underbrace{\exp\left(-\frac{\|X_i P_1 - X_j P_1\|_2^2}{2\sigma^2}\right)}_{K(X_i P_1, X_j P_1)}
\end{aligned}
\tag{8}
$$

The pairwise distances $\|X_i - X_j\|_2$ are invariant to the column permutation of $X$, because $P_1$ is a permutation matrix. Therefore, we can obtain $K_{XP_1} = K_X$.

Similarly, it is easily derived $K_{YP_2} = K_Y$ as follows,

$$
\begin{aligned}
k(Y_i, Y_j) &= \exp\left(-\frac{\|Y_i - Y_j\|_2^2}{2\sigma^2}\right) \\
&= \underbrace{\exp\left(-\frac{\|Y_i P_2 - Y_j P_2\|_2^2}{2\sigma^2}\right)}_{K(Y_i P_2, Y_j P_2)}
\end{aligned}
\tag{9}
$$

In this way, we have

$$
\begin{aligned}
\text{HSIC}(X, Y) &= \frac{1}{(m-1)^2}\text{tr}(K_X H K_Y H) \\
&= \underbrace{\frac{1}{(m-1)^2}\text{tr}(K_{XP_1} H K_Y H)}_{\text{HSIC}(XP_1, Y)} \\
&= \underbrace{\frac{1}{(m-1)^2}\text{tr}(K_X H K_{YP_2} H)}_{\text{HSIC}(X, YP_2)} \\
&= \underbrace{\frac{1}{(m-1)^2}\text{tr}(K_{XP_1} H K_{YP_2} H)}_{\text{HSIC}(XP_1, YP_2)}
\end{aligned}
\tag{10}
$$

Taking Eq.10 into Eq. 1, we have

$$
\begin{aligned}
\mathrm{CKA}(X, Y) &= \frac{\mathrm{HSIC}(X, Y)}{\sqrt{\mathrm{HSIC}(X, X) \cdot \mathrm{HSIC}(Y, Y)}} \\
&= \underbrace{\frac{\mathrm{HSIC}(XP_1, Y)}{\sqrt{\mathrm{HSIC}(XP_1, XP_1) \cdot \mathrm{HSIC}(Y, Y)}}}_{\mathrm{CKA}(XP_1, Y)} \\
&= \underbrace{\frac{\mathrm{HSIC}(X, YP_2)}{\sqrt{\mathrm{HSIC}(X, X) \cdot \mathrm{HSIC}(YP_2, YP_2)}}}_{\mathrm{CKA}(X, YP_2)} \\
&= \underbrace{\frac{\mathrm{HSIC}(XP_1, YP_2)}{\sqrt{\mathrm{HSIC}(XP_1, XP_1) \cdot \mathrm{HSIC}(YP_2, YP_2)}}}_{\mathrm{CKA}(XP_1, YP_2)}
\end{aligned}
\tag{11}
$$

Finally, we obtain

$$
\mathrm{CKA}(X, Y) = \mathrm{CKA}(XP_1, Y) = \mathrm{CKA}(X, YP_2) = \mathrm{CKA}(XP_1, YP_2)
\tag{12}
$$

### A.2  CASE 2: SCALING INVARIANCE

**For Linear CKA**, let $\tilde{X} = c_1 X$ and $c_1 \in \mathbb{R}^+$. Then,

$$
\begin{aligned}
K_{\tilde{X}} &= \tilde{X}\tilde{X}^\top \\
&= (c_1 X)(c_1 X)^\top \\
&= c_1^2 X X^\top \\
&= c_1^2 K_X
\end{aligned}
\tag{13}
$$

Similarly, let $\tilde{Y} = c_2 Y$ and $c_2 \in \mathbb{R}^+$. Then,

$$
\begin{aligned}
K_{\tilde{Y}} &= \tilde{Y}\tilde{Y}^\top \\
&= (c_2 Y)(c_2 Y)^\top \\
&= c_2^2 Y Y^\top \\
&= c_2^2 K_Y.
\end{aligned}
\tag{14}
$$

In this way,

$$
\begin{aligned}
\mathrm{HSIC}(c_1 X, c_2 Y) &= \frac{1}{(m-1)^2}\mathrm{tr}(K_{\tilde{X}} H K_{\tilde{Y}} H) \\
&= \frac{1}{(m-1)^2}\mathrm{tr}(c_1^2 K_X H c_2^2 K_Y H) \\
&= \frac{1}{(m-1)^2}\mathrm{tr}(c_1^2 c_2^2 K_X H K_Y H) \\
&= \frac{c_1^2 c_2^2}{(m-1)^2}\mathrm{tr}(K_X H K_Y H) \\
&= c_1^2 c_2^2 \mathrm{HSIC}(X, Y).
\end{aligned}
\tag{15}
$$

Accordingly,

$$\begin{aligned}
\text{HSIC}(c_1 X, c_1 X) &= \frac{1}{(m-1)^2} \text{tr}(K_{\tilde{X}} H K_{\tilde{X}} H) \\
&= \frac{1}{(m-1)^2} \text{tr}(c_1^2 K_X H c_1^2 K_X H) \\
&= \frac{1}{(m-1)^2} \text{tr}(c_1^4 K_X H K_X H) \\
&= \frac{c_1^4}{(m-1)^2} \text{tr}(K_X H K_X H) \\
&= c_1^4 \text{HSIC}(X, X).
\end{aligned} \tag{16}$$

$$\begin{aligned}
\text{HSIC}(c_2 Y, c_2 Y) &= \frac{1}{(m-1)^2} \text{tr}(K_{\tilde{Y}} H K_{\tilde{Y}} H) \\
&= \frac{1}{(m-1)^2} \text{tr}(c_2^2 K_Y H c_2^2 K_Y H) \\
&= \frac{1}{(m-1)^2} \text{tr}(c_2^4 K_Y H K_Y H) \\
&= \frac{c_2^4}{(m-1)^2} \text{tr}(K_Y H K_Y H) \\
&= c_2^4 \text{HSIC}(Y, Y).
\end{aligned} \tag{17}$$

Therefore, we have

$$\begin{aligned}
\text{CKA}(c_1 X, c_2 Y) &= \frac{\text{HSIC}(c_1 X, c_2 Y)}{\sqrt{\text{HSIC}(c_1 X, c_1 X) \cdot \text{HSIC}(c_2 Y, c_2 Y)}} \\
&= \frac{c_1^2 c_2^2 \text{HSIC}(X, Y)}{\sqrt{c_1^4 \text{HSIC}(X, X) \cdot c_2^4 \text{HSIC}(Y, Y)}} \\
&= \frac{c_1^2 c_2^2 \text{HSIC}(X, Y)}{c_1^2 c_2^2 \sqrt{\text{HSIC}(X, X) \cdot \text{HSIC}(Y, Y)}} \\
&= \underbrace{\frac{\text{HSIC}(X, Y)}{\sqrt{\text{HSIC}(X, X) \cdot \text{HSIC}(Y, Y)}}}_{\text{CKA}(X,Y)}
\end{aligned} \tag{18}$$

Finally, we obtain

$$\text{CKA}(X, Y) = \text{CKA}(c_1 X, c_2 Y) \tag{19}$$

**For RBF CKA**, the RBF kernel function is

$$\begin{aligned}
k(c_1 X_i, c_1 X_j) &= \exp\left(-\frac{\|c_1 X_i - c_1 X_j\|_2^2}{2\sigma^2}\right) \\
&= \exp\left(-\frac{c_1^2 \|X_i - X_j\|_2^2}{2\sigma^2}\right)
\end{aligned} \tag{20}$$

Following Kornblith et al. (2019), the bandwidth $\sigma$ is chosen as a fraction of the median distance, *i.e.*, $\sigma = \alpha \cdot \text{median}(\|X_i - X_j\|_2)$ for the constant $\alpha > 0$. In this way, Eq. 20 is transformed as,

$$k(c_1 X_i, c_1 X_j) = \exp\left(-\frac{c_1^2 \|X_i - X_j\|_2^2}{2(\alpha c_1^2 \cdot \mathrm{median}(\|X_i - X_j\|_2))^2}\right)$$
$$= \underbrace{\exp\left(-\frac{c_1^2 \|X_i - X_j\|_2^2}{2c_1^2 \sigma^2}\right)}_{k(X_i, X_j)}. \tag{21}$$

Similarly, it is easily derived $k(c_2 Y_i, c_2 Y_j) = k(Y_i, Y_j)$ as follows,

$$k(c_2 Y_i, c_2 Y_j) = \exp\left(-\frac{c_2^2 \|Y_i - Y_j\|_2^2}{2(\alpha c_2^2 \cdot \mathrm{median}(\|Y_i - Y_j\|_2))^2}\right)$$
$$= \underbrace{\exp\left(-\frac{c_2^2 \|Y_i - Y_j\|_2^2}{2c_2^2 \sigma^2}\right)}_{k(Y_i, Y_j)}. \tag{22}$$

Therefore, we can obtain $\mathrm{HSIC}(X, Y) = \mathrm{HSIC}(c_1 X, c_2 Y)$, $\mathrm{HSIC}(X, X) = \mathrm{HSIC}(c_1 X, c_1 X)$, and $\mathrm{HSIC}(Y, Y) = \mathrm{HSIC}(c_2 Y, c_2 Y)$

Finally, we have

$$\mathrm{CKA}(c_1 X, c_2 Y) = \frac{\mathrm{HSIC}(c_1 X, c_2 Y)}{\sqrt{\mathrm{HSIC}(c_1 X, c_1 X) \cdot \mathrm{HSIC}(c_2 Y, c_2 Y)}}$$
$$= \frac{\mathrm{HSIC}(X, Y)}{\sqrt{\mathrm{HSIC}(X, X) \cdot \mathrm{HSIC}(Y, Y)}}$$
$$= \mathrm{CKA}(X, Y). \tag{23}$$

Finally, we obtain
$$\mathrm{CKA}(X, Y) = \mathrm{CKA}(c_1 X, c_2 Y). \tag{24}$$

# B    THE EFFECTIVENESS OF CLASSIFIERS TRAINED ON REPRESENTATIONS OF A VICTIM MODEL

This appendix provides a detailed analysis of the experiments conducted to evaluate the effectiveness of classifiers trained on the representations of a victim model to identify whether a suspect model is derived from it, thus protecting its intellectual property. We explore the classifiers' accuracy when utilizing representations from different layers to train classifiers and applying them to the corresponding layers of the suspect model (B.1), as well as applying classifiers trained on one layer's representation to representations from other layers of the suspect model (B.2).

## B.1    APPLY CLASSIFIERS TO THE CORRESPONDING LAYER

Research has shown that representations from the middle and later layers of LLMs contain rich encoded information, which can be used to classify high-dimensional concepts, such as safety or unsafety, and honesty or dishonesty (Burns et al., 2023; Rimsky et al., 2023; Zou et al., 2023; Qian et al., 2024a;b; Chen et al., 2025). Following Section 3, we explore the effectiveness of classifiers trained on representations from different layers.

Specifically, we use Llama-2-7b and llama-2-13b as victim models, extracting representations from the 24th and 30th layers of Llama-2-7b and from the 32nd and 40th layers of Llama-2-13b for the TruthfulQA dataset. We then train various classifiers (*e.g.*, linear, MLP, CNN, GCN) on representations from each layer. These classifiers are subsequently applied to various suspect models, including LLMs derived from the victim models as well as unrelated LLMs.

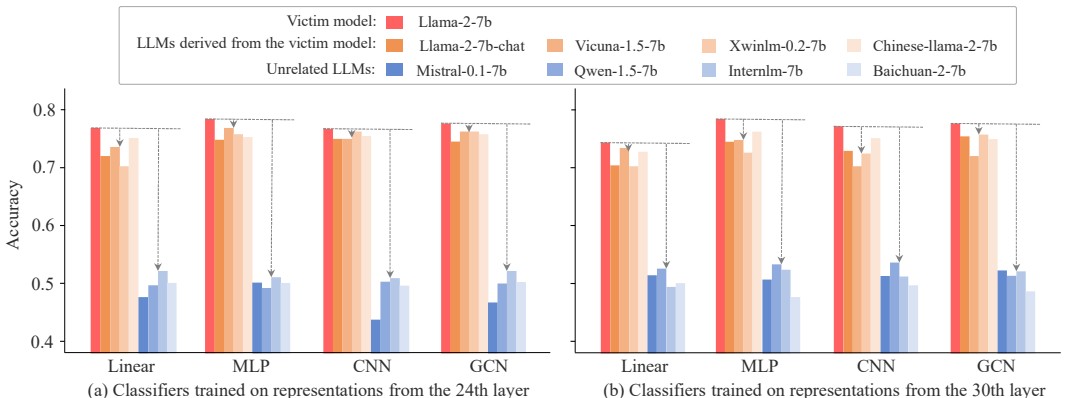

Figure 7: Accuracies of classifiers trained on representations from Llama-2-7b.

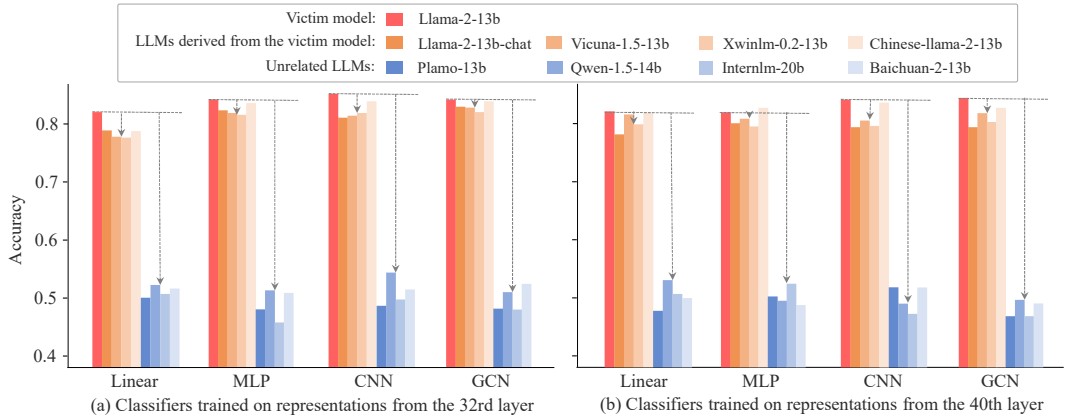

Figure 8: Accuracies of classifiers trained on representations from Llama-2-13b.

Classifiers trained on representations from different layers of the victim model are all capable of identifying whether a suspect model is derived from the victim model. Figures 7 and 8 show the results of applying classifiers trained on representations from the 24th and 30th layers of Llama-2-7b and from the 32nd and 40th layers of Llama-2-13b to suspect models on the TruthfulQA dataset. It can be observed that across different layers, all classifiers (linear, MLP, CNN, GCN) achieve an accuracy of over 70% on representations from LLMs derived from the victim model. This accuracy is close to the classification results of the victim model itself. However, the accuracy dropped to about 50% when applied to representations from unrelated models, which is close to random guessing and significantly lower than the classification results on the victim model's representations.

The results demonstrate that REEF, our representation-based fingerprinting method, does not depend on representations from any specific layer. By leveraging the powerful representation modeling capabilities of LLMs, REEF can use representations from various layers to identify the victim model within a suspect model, thereby protecting its intellectual property.

## B.2 APPLY CLASSIFIERS CROSS LAYERS

To further investigate the generalizability of our approach, we conduct cross-layer experiments by applying classifiers trained on representations from one layer to representations from other layers. For instance, we apply a linear classifier trained on the 18th layer representations of Llama-2-7b to the 24th layer representations of suspect models. This cross-layer analysis provides insights into the similarity of representations across different layers of the model.

Following the same training process as previously described, for Llama-2-7b, we select one layer's representations from the 18th, 24th, or 30th layer to train a linear classifier, which is then applied

Table 2: Accuracies of classifiers applied across layers for victim model Llama-2-7b. Gray shading indicates that the classifier was trained using representations from that specific layer.

| | Victim LLM | LLMs derived from the victim model | | | | Unrelated LLMs | | | |
| | Llama-2 -7b | Llama-2 -7b-chat | Vicuna-1.5 -7b | Chinese- llama-2-7b | Xwimlm -7b | Mistral -7b | Baichuan -2-7b | Qwen -1.5-7b | Internlm -7b |
|---|---|---|---|---|---|---|---|---|---|
| **Layer-18** | 0.8003 | 0.7437 | 0.7642 | 0.7578 | 0.7421 | 0.5078 | 0.4513 | 0.5063 | 0.5094 |
| **Layer-24** | 0.7123 | 0.7008 | 0.6965 | 0.7081 | 0.7060 | 0.4953 | 0.5314 | 0.5283 | 0.5016 |
| **Layer-30** | 0.6715 | 0.6778 | 0.6809 | 0.6762 | 0.6636 | 0.5031 | 0.4890 | 0.5094 | 0.5252 |
| **Layer-18** | 0.7014 | 0.7030 | 0.7124 | 0.7077 | 0.6967 | 0.4717 | 0.5283 | 0.5418 | 0.5130 |
| **Layer-24** | 0.7720 | 0.7233 | 0.7390 | 0.7055 | 0.7547 | 0.4780 | 0.4984 | 0.5235 | 0.5031 |
| **Layer-30** | 0.6723 | 0.6629 | 0.7085 | 0.6660 | 0.6975 | 0.4513 | 0.4953 | 0.5126 | 0.4764 |
| **Layer-18** | 0.6982 | 0.6945 | 0.6914 | 0.6950 | 0.6840 | 0.5225 | 0.5096 | 0.4827 | 0.5189 |
| **Layer-24** | 0.7097 | 0.7050 | 0.7191 | 0.7034 | 0.7233 | 0.5189 | 0.4959 | 0.4591 | 0.4686 |
| **Layer-30** | 0.7453 | 0.7061 | 0.7360 | 0.7045 | 0.7296 | 0.5157 | 0.5270 | 0.4953 | 0.5036 |

Table 3: Accuracies of classifiers applied across layers for victim model Llama-2-13b. Gray shading indicates that the classifier was trained using representations from that specific layer.

| | Victim model | LLMs derived from the victim model | | | | Unrelated LLMs | | | |
| | Llama-2 -13b | Llama-2 -13b-chat | Vicuna-1.5 -13b | Chinese- llama-2-13b | Xwimlm -13b | Plamo -13b | Baichuan -2-13b | Qwen -1.5-14b | Internlm -20b |
|---|---|---|---|---|---|---|---|---|---|
| **Layer-24** | 0.8412 | 0.8223 | 0.8066 | 0.8081 | 0.8223 | 0.4827 | 0.5283 | 0.4276 | 0.4946 |
| **Layer-32** | 0.8050 | 0.7783 | 0.7814 | 0.7909 | 0.8082 | 0.4811 | 0.4827 | 0.4450 | 0.4546 |
| **Layer-40** | 0.7767 | 0.7248 | 0.7783 | 0.7421 | 0.7594 | 0.4780 | 0.5372 | 0.4906 | 0.4289 |
| **Layer-24** | 0.8381 | 0.7925 | 0.8113 | 0.8145 | 0.8192 | 0.4874 | 0.5329 | 0.5236 | 0.4996 |
| **Layer-32** | 0.8223 | 0.7909 | 0.7799 | 0.7799 | 0.7909 | 0.5000 | 0.5220 | 0.5079 | 0.5057 |
| **Layer-40** | 0.7767 | 0.7484 | 0.7642 | 0.7186 | 0.7767 | 0.5083 | 0.5152 | 0.5350 | 0.4893 |
| **Layer-24** | 0.8302 | 0.827 | 0.8129 | 0.8113 | 0.8223 | 0.4858 | 0.5412 | 0.5000 | 0.4734 |
| **Layer-32** | 0.8113 | 0.7783 | 0.8035 | 0.7814 | 0.8003 | 0.4560 | 0.5397 | 0.5031 | 0.4896 |
| **Layer-40** | 0.8239 | 0.7842 | 0.8187 | 0.8014 | 0.8207 | 0.4780 | 0.5314 | 0.5173 | 0.5000 |

to the representations from the other two layers across various suspect models. For instance, linear classifiers trained on representations from the 18th layer are applied to the representations of the 24th and 30th layers in different suspect models. Similarly, for Llama-2-13b, we choose representations from the 24th, 32nd, or 40th layer to conduct the same cross-layer classifier application. The experimental results are presented in Tables 2 and 3, respectively, which provide detailed accuracy metrics for each cross-layer classification task.

Table 2 shows that the classifier trained on the specific layer's representations (*e.g.*, 18th layer) of Llama-2-7b, when applied to other layers' representations (*e.g.*, 24th and 30th layer) of suspect models, maintained the accuracy 70% for derived models and 50% for unrelated models. Table 3 demonstrates similar results for experiments conducted on the larger Llama-2-13b model, with significantly distinct accuracy ranges. These results indicate that classifiers trained on one layer's representations remain effective when applied to other layers, suggesting a significant similarity in the representation spaces across different layers of the model.

The ability of these classifiers to generalize across layers further strengthens the reliability of our fingerprinting detection method. It indicates that the distinctive features learned by the classifiers are not confined to a specific layer but are present throughout the model's architecture. This characteristic enhances the robustness of our approach, making the use of representations as fingerprints for protecting the intellectual property of the victim model more reliable through cross-layer validation. Additionally, this insight inspires us to use heatmaps to depict the CKA similarity between the representations of the victim LLM and those of various suspect LLMs across the same samples, as presented in the main text.

## C    HEATMAPS OF THE VICTIM MODEL AND DIFFERENT SUSPECT MODELS

In Section 5.2, we report REEF's similarity of representations from the 18th layer between the victim model and various suspect models. These suspect models are derived from the victim model through a range of developments, including fine-tuning, pruning, merging, permutation, and scaling transformation. To provide a clearer and more intuitive comparison, we supplement this analysis with heatmaps in Figure 9, depicting the layer-wise and inter-layer CKA similarity of representations for the same samples between each pair of victim and suspect models. Figure 9 demonstrates that, regardless of the type of development applied to the victim model, our representation-based fingerprint REEF can significantly identify the victim model, as shown by the high CKA similarities in the heatmap.

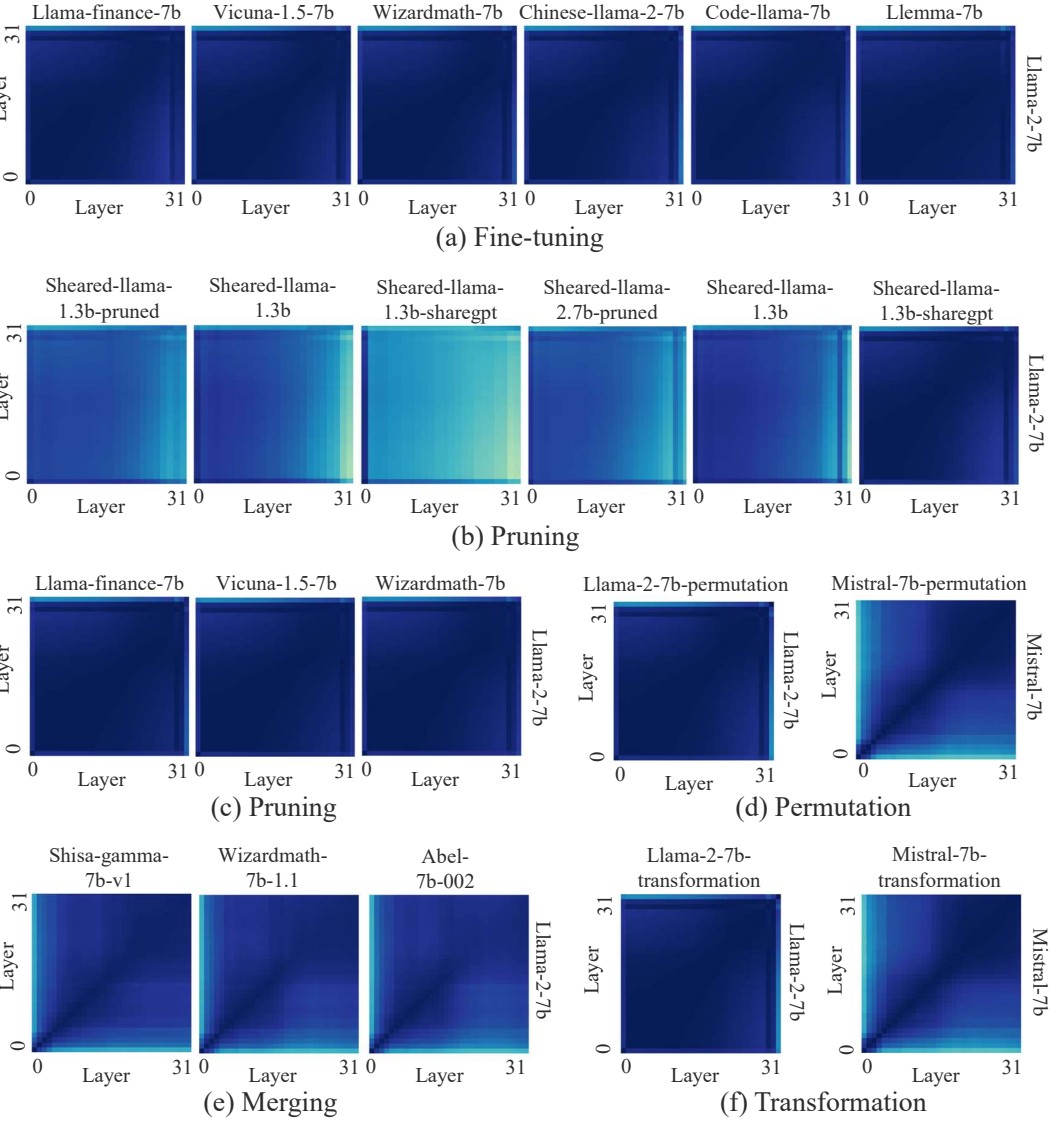

Figure 9: Heatmaps depicting the layer-wise and inter-layer CKA similarity of representations for the same samples between each pair of victim and suspect models.

Table 4: Similarity of various LLM fingerprinting methods applied to suspect models developed from the Qwen-2.5-7b.

|  | Qwen-2.5-7b-coder | Qwen-2.5-7b-pruning | Qwent-7b | Qwen-2.5-7b-permutation |
|---|---|---|---|---|
| **PCS** | 0.6769 | 0.0000 | 0.9499 | 0.0000 |
| **ICS** | 0.9461 | 0.7638 | 0.9989 | 0.9197 |
| **Logits** | 0.0670 | 0.9999 | 0.8167 | 0.0000 |
| **REEF** | 0.9411 | 0.9785 | 0.9599 | 1.0000 |

Table 5: Similarity of various LLM fingerprinting methods applied to suspect models developed from the Mistral-7b.

|  | Mathstral-7B | Mistral-7b-pruning | Evollm-jp-7b | Mistral-7b-permutation |
|---|---|---|---|---|
| **PCS** | 0.9803 | 0.0000 | 0.9989 | 0.0000 |
| **ICS** | 0.9883 | 0.6392 | 0.9928 | 0.9847 |
| **Logits** | 0.3867 | 0.9999 | 0.9999 | 0.0000 |
| **REEF** | 0.9344 | 0.9868 | 0.9516 | 1.0000 |

## D    REEF'S APPLICATION ACROSS DIFFERENT LLM FAMILIES

To demonstrate the generalizability of REEF across different model families, we select Qwen-2.5-7b and Mistral-7b as victim models. Then, We apply REEF to various suspect models derived from the victim model, including fine-tuning, pruning, merging, and parameter perturbation.

For the Qwen-2.5-7b victim model, we use several variants through different modification approaches: domain-specific fine-tuning with code data (Qwen-2.5-7b-coder), 20% block-wise pruning (Qwen-2.5-7b-pruning), weight merging between qwen-2-7b and qwen-2.5-7b (Qwent-7b), and parameter perturbation (Qwen-2.5-7b-permutation). Similarly, for the Mistral-7b victim model, we use variants including mathematical domain fine-tuning (Mathstral-7B), 20% block-wise pruning (Mistral-7b-pruning), a weighted parameter merge of Shisa-gamma-7b-v1, Wizardmath-7b-1.1, and Abel-7b-002 (Evollm-jp-7b), and parameter perturbation (Mistral-7b-permutation).

As shown in Tables 4 and 5, REEF consistently achieves high CKA similarity scores across all suspect models and victim models. This demonstrates that REEF can effectively identify the victim model regardless of whether it is Qwen-2.5-7b or Mistral-7b, even after various downstream modifications. The robust performance across different LLM families underscores the general effectiveness of our approach.

## E    EVADING REEF WITH FINE-TUNING

We hypothesize that malicious developers aware of the REEF approach might attempt to design customized loss functions during fine-tuning to evade detection. Given that REEF determines model similarity based on the representation similarity between the suspect and victim models, malicious developers aiming to avoid detection would likely design their customized loss to maximize the representational divergence between these models.

Based on this premise, we designed two experiments to attempt to circumvent REEF detection:

- Integrating the task loss with a customized loss during the fine-tuning process, aiming to achieve the fine-tuning objective while maximizing the representational dissimilarity with the victim model.

- Fine-tuning the victim model solely using the customized loss, attempting to maximize the representational dissimilarity between the original and fine-tuned models.

To evaluate these scenarios, we conduct experiments using the OPT-1.3B model (Zhang et al., 2022) and the E2E NLG Challenge dataset (Novikova et al., 2017) for fine-tuning. We employ the LoRA technique (Hu et al., 2021) for efficient adaptation. The customized loss is designed to measure the CKA similarity between the logits of the original and fine-tuned models.

For the first scenario, we formulate a combined loss function: $\mathcal{L} = \mathcal{L}_{\text{task}} + \lambda\mathcal{L}_{\text{custom}}$, where $\mathcal{L}_{\text{task}}$ is the task-specific loss (e.g., cross-entropy for the E2E NLG Challenge), $\mathcal{L}_{\text{custom}}$ is designed to adjust the CKA similarity between the original and fine-tuned models, and $\lambda$ is the weighting coefficient. As for $\mathcal{L}_{\text{custom}}$, we design two types of loss functions. One is the direct CKA similarity between the logits of the original and fine-tuned models, namely CKA loss. Specifically, the customized CKA loss is calculated using Equation 1, that is:

$$\text{CKA}(\text{LG}_{\text{ori}}, \text{LG}_{\text{ft}}) = \frac{\text{HSIC}(\text{LG}_{\text{ori}}, \text{LG}_{\text{ft}})}{\sqrt{\text{HSIC}(\text{LG}_{\text{ori}}, \text{LG}_{\text{ori}}) \cdot \text{HSIC}(\text{LG}_{\text{ft}}, \text{LG}_{\text{ft}})}}, \tag{25}$$

where $\text{LG}_{\text{ori}}$ and $\text{LG}_{\text{ft}}$ represent the logits of the original and fine-tuned models on the same sample.

The other is the Wasserstein loss, which is used to maximize the divergence between the logits of the original and fine-tuned models, defined as $\mathcal{L}_{\text{W}} = \max\left(\mathbb{E}_{x \sim \mathcal{D}}\left[W(\text{LG}_{\text{ori}}(x), \text{LG}_{\text{ft}}(x))\right]\right)$, where $W(\cdot, \cdot)$ represents the Wasserstein distance between two distributions (e.g., logits of the original and fine-tuned models)

In this scenario, incorporating different weighting coefficients for the customized loss during the combined fine-tuning process failed to reduce the representational similarity between the fine-tuned model and the original model. This suggests that during fine-tuning, the model continues to rely on the representation modeling capabilities of the original language model. Consequently, achieving ECE task objectives necessarily preserves the representational distribution.

In the second scenario, although targeted fine-tuning can increase the distributional divergence in the representation space between the suspect and victim models, the suspect model loses its fundamental language expression capabilities, rendering its outputs meaningless. For example, the fine-tuned model may only respond with repetitive patterns such as "and and and and ..." for any input, demonstrating a complete loss of linguistic coherence and utility.

Therefore, our method demonstrates resilience against malicious actors' attempts to evade detection through fine-tuning strategies. These findings underscore the robustness of REEF in identifying the victim model, even in the face of sophisticated evasion techniques.

## F  REEF EVALUATION ON INDEPENDENTLY TRAINED MODELS WITH SIMILAR DATASETS

To evaluate the performance of REEF on models independently trained on similar datasets, we perform pre-training from scratch using the 1.5-Pints pre-training corpus, *i.e.*, Expository-Prose-V1 (Tan & Wang, 2024). A new model is locally trained with varying data orders and hyperparameter configurations, such as learning rates and batch sizes.

Specifically, 1.5-Pints is a Large Language Model that emphasizes data quality over quantity in LLM training, featuring a meticulously curated pre-training corpus of 57 billion tokens. Using the dataset provided in the original paper, we conduct pre-training on 8 A100 GPUs with different random seeds for data shuffling. The hyperparameters for pre-training are set as follows: a global batch size of 512, a learning rate of 4e-4, a micro-batch size of 8, a maximum of 56,960 steps, a weight decay of 0.1, beta1 of 0.9, beta2 of 0.95, gradient clipping at 1.0, and a minimum learning rate of 4e-5. The pre-trained model undergoes supervised fine-tuning to obtain 1.5-pints-sft, followed by safety alignment to generate 1.5-pints-dpo.

In our experimental setup, we choose 1.5-pints-dpo as the suspect model, which is obtained by conducting further safety alignment on the 1.5-pints-sft model. We perform REEF on 1.5-pints-dpo with 1.5-pints-sft and 1.5-pints-2k to test whether REEF can accurately identify its source from models trained independently on the same dataset. The performance of REEF across these two victim models is illustrated in Table 6.

REEF can still correctly identify the victim models from models that are independently trained on the same dataset. As shown in Table 6, the CKA similarity highlights differences between 1.5-pints-sft and its derived model (1.5-pints-dpo), compared to models pre-trained on the same dataset

Table 6: The CKA similarity of 1.5-pints-dpo with 1.5-pints-sft and 1.5-pints-2k, respectively.

|  | 8th Layer | 12th Layer | 16th Layer | 20th Layer |
|---|---|---|---|---|
| **1.5-pints-sft** | 0.9983 | 0.9978 | 0.9908 | 0.9884 |
| **1.5-pints-2k** | 0.7632 | 0.7603 | 0.7723 | 0.7931 |

Table 7: General capability evaluation of 1.5-Pints model variants.

|  | ARC | RACE | MatQA | BoolQ | ToxiGen | WinoGrande | Lambada | PPL |
|---|---|---|---|---|---|---|---|---|
| **1.5-Pints-2k** | 0.4727 | 0.3292 | 0.2452 | 0.5229 | 0.4245 | 0.5383 | 0.4751 | 12.52 |
| **1.5-Pints-ft** | 0.4842 | 0.334 | 0.2536 | 0.4498 | 0.4085 | 0.5335 | 0.4508 | 16.18 |
| **1.5-Pints-dpo** | 0.4822 | 0.3464 | 0.2506 | 0.5391 | 0.4064 | 0.5233 | 0.4485 | 16.83 |

with varied data orders and hyperparameters (1.5-pints-2k). This discriminative ability of REEF minimizes false positives when analyzing models independently trained on identical datasets.

Furthermore, our comprehensive evaluation comparing independently pre-trained models (1.5-pints-sft and 1.5-pints-dpo) with the original paper's models (1.5-pints-2k) across multiple datasets demonstrates consistent and reliable general capabilities, as shown in Table 7.

## G    LIMITATIONS

There are several limitations to this work. Firstly, our study focuses on open-source LLMs, which allows model owners and third parties (e.g., regulatory authorities) to verify and protect model ownership. However, for closed-source models, the lack of access to their representations limits the applicability of our approach. Secondly, regarding fine-tuning, due to the high cost of fine-tuning with extensive data (more than 700B), although we discuss the effectiveness of our method in main paper, empirical validation is lacking.

## H    FUTURE WORK

While REEF demonstrates robust performance in identifying root victim models, there are several promising directions for future research. A key restriction of our current approach is that REEF primarily focuses on direct lineage identification between suspect models and their root origins, rather than tracking multi-generational model development paths. Future work could explore hybrid approaches that combine our fingerprinting technique with watermarking methods to enable comprehensive model genealogy tracking. This would allow for not only identifying the root origin but also verifying the complete development pathway of suspect models through multiple generations of modifications, including fine-tuning, merging, and other adaptations. Such capabilities would be particularly valuable as the LLM ecosystem becomes increasingly complex with models being iteratively developed and modified across different organizations.

