# OpenReview forum: "REEF: Representation Encoding Fingerprints for Large Language Models"
_ICLR.cc/2025/Conference — ICLR 2025 Oral_

### Official Review · Reviewer_HD36 · 2024-10-30

**Soundness:** 4
**Presentation:** 3
**Contribution:** 3
**Rating:** 8
**Confidence:** 4

**Summary:**

This paper proposes a method to identify descendant models of a large language model by using a kernel-based similarity measure between the descendants and the original model. This similarity measure enables robust similarity computation and effective detection of descendant models. However, the proposed method is only effective in a white-box scenario where the parameters of both the potential descendant models and the original model are publicly accessible. The detection performance using this similarity calculation is high and robust.

**Strengths:**

- S1: The paper is well-organized and easy to follow.
- S2: The proposed method identifies descendant models of a large language model through kernel-based similarity with the original model, allowing for robust and effective discovery of descendant models.
- S3: The method is intuitive and well-supported by analytical study.

**Weaknesses:**

- W1: The security model and fingerprint verification setting are not clearly defined. The proposed method is only effective in a white-box scenario where the parameters of both the descendant models and the original model are publicly accessible. To avoid misunderstandings regarding potential applications in black-box scenarios, such as through API access, the paper should clarify the security model, including the verifier’s capabilities.
- W2: The proposed fingerprint verification method is limited to direct descendant models of the original “root” model and is not designed for verifying claims from these descendants against further generations. This limitation is significant for the proposed fingerprint verification approach.

**Questions:**

Address W1.

**Details Of Ethics Concerns:**

Nothing.

---

> ### Author Response · Authors · 2024-11-24
> **Response to Reviewer HD36**
>
> Thank you for your great efforts in the review of this paper. We will try our best to answer all your questions. Please let us know if you still have further concerns, or if you are not satisfied with the current responses, so that we can further update the response ASAP.
>
> ---
>
> **Q1:** "The security model and fingerprint verification setting are not clearly defined. The proposed method is only effective in a white-box scenario where the parameters of both the descendant models and the original model are publicly accessible. To avoid misunderstandings regarding potential applications in black-box scenarios, such as through API access, the paper should clarify the security model, including the verifier’s capabilities."
>
> **A1:** Thank you for your valuable suggestions. We **have followed your suggestions** to clearly define the security model and fingerprint verification setting and revise the paper in Lines 089, 132-133, 185, and 535-536 of the updated version of manuscript.
>
> * REEF performs LLM fingerprinting under the open-source LLMs' scenario, because REEF needs to use the feature representations from the inference process of LLMs. In this way, REEF is only effective in a white-box scenario (open-source LLMs) and cannot be applied to black-box scenarios, such as through API access.
> * REEF is utilized to identify "whether the suspect model is a subsequent development of the victim model (e.g., Code-llama trained from Llama-2) or is developed from scratch (e.g., Mistral)" (Lines 30-32).
>
> REEF verification details:
>
> 1. First, the process begins with extracting feature representations of the same samples (e.g., 200 samples from TruthfulQA as used in this paper) from model layers.
> 2. Then, the CKA method (Lines 184-194) is employed to calculate the similarity between the representation of the suspect model and that of the victim model.
> 3. Finally, if the CKA similarity exceeds a certain threshold (e.g., 0.8, as shown in the green region of Table 1), the suspect model is identified to be derived from the victim model; otherwise, it is not.
>
> ---
>
> **Q2:** "The proposed fingerprint verification method is limited to direct descendant models of the original 'root' model and is not designed for verifying claims from these descendants against further generations. "
>
> **A2:** Thank you. We acknowledge that REEF is designed to identify which victim model serves as the root origin of a suspect model and is not suitable for determining whether the suspect model is further developed from its descendant models. The latter one is also interesting and usually solved by watermark methods, which is beyond the scope of this paper. We look forward to exploring it in future research.

---

> > ### Comment · Reviewer_HD36 · 2024-11-26
> >
> > Thank you for your detailed response.
> >
> > > A1: Thank you for your valuable suggestions. We have followed your suggestions to clearly define the security model and fingerprint verification setting and revise the paper in Lines 089, 132-133, 185, and 535-536 of the updated version of manuscript.
> >
> > Your response adequately addresses my concerns regarding the security model.
> >
> >
> > > A2: Thank you. We acknowledge that REEF is designed to identify which victim model serves as the root origin of a suspect model and is not suitable for determining whether the suspect model is further developed from its descendant models. The latter one is also interesting and usually solved by watermark methods, which is beyond the scope of this paper. We look forward to exploring it in future research.
> >
> > I recommend explicitly stating in the main text the limitation that the proposed method is designed to identify which victim model serves as the root origin. Including this clarification will help better define the scope and target applications of your work, thereby enhancing its motivation and focus.

---

> > > ### Author Response · Authors · 2024-11-26
> > >
> > > Thank you for your feedback. We have **followed your suggestion and clarified** that "the proposed method is designed to identify which victim model serves as the root origin" in Lines 31, 80-81, 89, 187, 247, and 536-537 of the revised manuscript. We have also included this as future work in Appendix H.
> > >
> > > Please feel free to share any other insightful suggestions. We would like to improve the paper with any valuable comments.

---

> > > > ### Comment · Reviewer_HD36 · 2024-11-27
> > > >
> > > > Thank you for your response.
> > > > All my concerns have been adequately addressed, and I will increase my rating to 8 (accept).

---

> > > > > ### Author Response · Authors · 2024-11-27
> > > > > **Thanks for your reply!**
> > > > >
> > > > > We would like to thank the reviewer for all the valuable comments and questions. We are grateful for your engagement in the rebuttal process.

---

### Official Review · Reviewer_Gx6d · 2024-11-03

**Soundness:** 2
**Presentation:** 3
**Contribution:** 3
**Rating:** 6
**Confidence:** 2

**Summary:**

The paper proposes to use CKA to measure the independence between two representations of two LLMs as the model fingerprints. Proposed issue is aiming to address the challenges brought by model pruning, model merging, and other techniques that can change the model architecture to make prior fingerprint detection useless. The experiments show the effectiveness of the proposed method.

**Strengths:**

- Protection of the IP of the large language model is an urgent and important topic.

- The experiments are extensive to show the effectiveness of the proposed method.

- The authors clearly identify the challenges and limitations of existing fingerprint methods.

**Weaknesses:**

The novelty is limited in replacing similarity measures of the prior work. There is a lack of theoretical analysis of proposed methods. Additionally, I'm confused about the first heatmap of Figure 6. It shows that the two LLMs trained on different datasets will have a high CKA similarity, which will lead the REEF to classify them into the same model. This means all models using the same architecture but trained by different datasets will share the same fingerprint. It doesn't make sense.

**Questions:**

Please refer to the weakness part.

---

> ### Author Response · Authors · 2024-11-24
> **Response to Reviewer Gx6d**
>
> Thank you for your great efforts in the review of this paper. We will try our best to answer all your questions. Please let us know if you still have further concerns, or if you are not satisfied with the current responses, so that we can further update the response ASAP.
>
> ---
>
> **Q1:** "The novelty is limited in replacing similarity measures of the prior work."
>
> **A1:** Thank you. The contribution of this paper is far extended beyond merely replacing the similarity measures from prior work. This paper proposes **a novel feature perspective of LLM fingerprints, which is simple and effective** as appreciated by Reviewers 4mRo and i9nF.
>
> **Protecting LLMs' intellectual properties (IPs) are nontrivial and challenging** . LLMs' IPs are very important for model owners and third parties, because the training costs of LLMs are extremely expensive. Watermarking techniques are classical methods for protecting IPs, but suffer from extra training costs, decreasing the model’s general capabilities, and being sensitive to subsequent fine-tuning and pruning operations.
>
> **REEF is a novel and new perspective on LLMs' representation-based fingerprints** . Specifically, REEF utilizes intrinsic feature representations to protect LLMs' IPs and it is training-free, maintaining general capabilities, and robust to various subsequent developments. Therefore, REEF is a **fundamentally new solution** to LLMs' fingerprints.
>
> ---
>
> **Q2:** "There is a lack of theoretical analysis of proposed methods."
>
> **A2:** Thank you. This paper has already included theoretical analysis in Lines 213-242 and Lines 810-1058. Specifically, Theorem 1 (Lines 213-235) theoretically proves that CKA is invariant to column-wise permutations and scaling transformations. Detailed derivations are provided in Appendix A (Lines 810-1058), which support the robustness of REEF. Consequently, Reviewer 4mRo comments that this paper "has already considered all important studies in its theoretical and empirical analyses."
>
> ---
>
> **Q3:** "I'm confused about the first heatmap of Figure 6. It shows that the two LLMs trained on different datasets will have a high CKA similarity, which will lead the REEF to classify them into the same model. This means all models using the same architecture but trained by different datasets will share the same fingerprint. It doesn't make sense. "
>
> **A3:** Thank you for your constructive suggestions. We acknowledge that Figure 6 may lead to misunderstanding.  We have **followed your suggestions** to clarify and revise Figure 6 in the updated version of manuscript.
>
> Figure 6 (a) indicates that the layer-wise CKA similarity between **Llama-2-7b itself** is almost 1, i.e., an LLM trained on the same dataset.
>
> In contrast, Figure 6 (b) indicates LLMs with the same architecture but trained on different datasets have low CKA feature similarity between each other. The four heatmaps are similar to the results of the victim model and unrelated LLMs in Figure 3 (Lines 216-232). Therefore, REEF can distinguish between models with the same architecture but different pre-training data.

---

> > ### Comment · Reviewer_Gx6d · 2024-11-26
> >
> > Thanks for your efforts, now I have no problem. I will increase my rating to 6.

---

> > ### Author Response · Authors · 2024-11-28
> > **Reminder: Revised Manuscript**
> >
> > Thanks for your constructive comments! In the latest revised manuscript, we have followed your suggestions and made revisions to Figure 6 and its corresponding discussion (Lines 486-494, 515-519). We have also highlighted the theoretical sections of the paper to improve clarity (Lines 212-215, 234-235, 810-1056). Once again, we appreciate the reviewer’s comments, which have helped us improve the paper.

---

> ### Author Response · Authors · 2024-11-26
> **Thanks for your reply!**
>
> We would like to thank the reviewer for all the valuable comments and questions. We are grateful for your engagement in the rebuttal process.
>
> Please feel free to share any other insightful suggestions. We would like to improve the paper with any valuable comments.

---

### Official Review · Reviewer_i9nF · 2024-11-04

**Soundness:** 3
**Presentation:** 3
**Contribution:** 3
**Rating:** 8
**Confidence:** 3

**Summary:**

This paper proposes REEF, a scheme for identifying “suspect” models that were derived from “victim” models. REEF does not require modifying the training process at all. The key idea of REEF is to measure the similarity between the feature representations of the two models. They first consider training a classifier to do so, however they observe that this approach is not robust. Their actual scheme instead computes a more robust similarity metric, called the centered kernel alignment, between the two models. This metric still captures similarity of the feature representations, but is robust to permutations. This work includes a comprehensive suite of experiments showing that REEF is robust to standard attacks such as permutation and pruning attacks.

**Strengths:**

This paper appears to be original, taking the known idea that models develop distinct feature representations, and introducing a novel use of the central kernel alignment score to measure similarity between these representations robustly. The scheme does not require changing the training algorithm of the original model, which is a nice property.

This paper is clearly written and well organized. The experiments are comprehensive in comparing the REEF scores of several known models, considering various practical attacks. The experiments also compare REEF to several existing fingerprinting schemes.

The result appears to be a significant incremental improvement. That is, REEF is a slightly more robust fingerprinting scheme than existing schemes. It does not introduce profoundly new ideas, but has solid practical improvement.

**Weaknesses:**

The biggest weakness is that REEF does not offer any provable false positive guarantee. Furthermore, this work does not measure the false positive rate, and indeed it is challenging to measure the false positive rate in a meaningful way since real-world LLMs are so expensive to train. Although this work does compare the REEF scores of several models, this sample size is not enough to extrapolate a false positive rate in general. For example, it is unclear how REEF would perform on two models that are independently trained on similar datasets– would REEF assert that one model is stolen from the other? The false positive rate is especially important if one wants REEF scores to be used to accuse a party of stealing a victim model.

It would also be interesting to see what happens when a model is fine-tuned to minimize its REEF score with the model it is derived from.

**Questions:**

How does REEF perform on models that were independently trained on the same (possibly publicly available) dataset? Is there a risk of false positives here?

How does one measure the false positive rate in general?

---

> ### Author Response · Authors · 2024-11-24
> **Response to Reviewer i9nF (Part Ⅰ)**
>
> Thank you for your great efforts in reviewing this paper and for your recognition of our work. We will try our best to answer all your questions. Please let us know if you still have further concerns, or if you are not satisfied with the current responses, so that we can further update the response ASAP.
>
> ---
>
> **Q1:** "How does REEF perform on models that were independently trained on the same (possibly publicly available) dataset? Is there a risk of false positives here?"
>
> **A1:** Thank you for your insightful suggestions. We **have followed your suggestions to conduct new experiments** to examine the performance of REEF on models independently trained on the same dataset.
>
> Due to the complexity and opacity of pre-training, it is almost impossible to reproduce an identical pre-training process (same data order, same hyperparameters). We are here to **independently train a new model on the same dataset but under different data orders and hyperparameter settings (e.g., learning rate, batch size.)**.
>
> Specifically, we pre-trained a 1.5B model ( **1.5-pints-sft** ) on the Expository-Prose-V1 dataset, which is used by **1.5-pints-2k** LLM [c1]. Thus, we have two models that are independently trained on the same dataset as victim models to explore the risk of false positives.
>
> Then, we choose 1.5-pints-dpo as the suspect model, which is obtained by conducting further safety alignment on the 1.5-pints-sft model. We perform REEF on 1.5-pints-dpo with 1.5-pints-sft and 1.5-pints-2k to test whether REEF can accurately identify its source from models trained independently on the same dataset. The following table indicates that **REEF can still correctly identify the victim models from models that are independently trained on the same dataset, without false positives.**
>
> |  | 8th Layer | 12th Layer | 16th Layer | 20th Layer |
> | --------------- | ----------- | ------------ | ------------ | ------------ |
> | 1.5-pints-sft | 0.9983    | 0.9978     | 0.9908     | 0.9884     |
> | 1.5-pints-2k  | 0.7632    | 0.7603     | 0.7723     | 0.7931     |
>
> Furthermore, the table below presents our evaluation results for independently pre-trained models (1.5-pints-sft and 1.5-pints-dpo) compared to the models provided in the paper (1.5-pints-2k) across several datasets, demonstrating the models' reliable general capabilities.
>
> |  | ARC    | RACE   | MathQA | BoolQ  | ToxiGen | WinoGrande | Lambada | PPL   |
> | --------------- | -------- | -------- | -------- | -------- | --------- | ------------ | --------- | ------- |
> | 1.5-pints-2k  | 0.4727 | 0.3292 | 0.2452 | 0.5229 | 0.4245  | 0.5383     | 0.4751  | 12.52 |
> | 1.5-pints-ft  | 0.4842 | 0.334  | 0.2536 | 0.4498 | 0.4085  | 0.5335     | 0.4508  | 16.18 |
> | 1.5-pints-dpo | 0.4822 | 0.3464 | 0.2506 | 0.5391 | 0.4064  | 0.5233     | 0.4485  | 16.83 |
>
> The above experiments and discussions are added in Appendix F.
>
> [c1] Tan, Calvin, and Jerome Wang. 1.5-Pints Technical Report: Pretraining in Days, Not Months--Your Language Model Thrives on Quality Data. *arXiv preprint arXiv:2408.03506* (2024).

---

> ### Author Response · Authors · 2024-11-24
> **Response to Reviewer i9nF (Part Ⅱ)**
>
> **Q2:** "How does one measure the false positive rate in general? "
>
> **A2:** Thank you for your constructive suggestions. We define victim model identification using REEF as a binary classification task, where the goal is to correctly identify whether the suspect model is derived from the victim model. We consider CKA similarity scores above 0.5 as indicating that the suspect model originates from the victim model (Lines 266-167, 324-327 in the updated version of manuscript). REEF can identify the victim models among the **30 suspect models** in Table 1 (Lines 324-352).
>
> Additionally, we conduct new experiments where Qwen-2.5-7b and Mistral-7barevictim models. Experimental results of **8 new suspect models** in the following table illustrate REEF's robustness across various scenarios including fine-tuning, pruning, merging, and parameter perturbation on these distinct model families.
>
> * Qwen-2.5-7b as the victim model.
>
> | | Qwen-2.5-7b-coder | Qwen-2.5-7b-pruning | Qwent-7b | Qwen-2.5-7b-permutation |
> | -------- | ------------------- | --------------------- | ---------- | ------------------------- |
> | PCS    | 0.6769            | 0.0000              | 0.9499   | 0.0000          |
> | ICS    | 0.9461            | 0.7638              | 0.9989   | 0.9197         |
> | Logits | 0.0670            | 0.9999              | 0.8167   | 0.0000        |
> | REEF   | 0.9411            | 0.9785              | 0.9599   | 1.0000    |
>
> * Mistral-7b as the victim model.
>
> |  | Mathstral-7B | Mistral-7b-pruning | Evollm-jp-7b | Mistral-7b-permutation |
> | -------- | ------- | ---------- | --------- | ----------- |
> | PCS    | 0.9803       | 0.0000             | 0.9989       | 0.0000                 |
> | ICS    | 0.9883       | 0.6392             | 0.9928       | 0.9847                 |
> | Logits | 0.3867       | 0.9999             | 0.9999       | 0.0000                 |
> | REEF   | 0.9344       | 0.9868             | 0.9516       | 1.0000                 |
>
> According to the victim model identification results of REEF among the 38 models mentioned, the following confusion matrix indicates that  **the false positive rate among the measured 38 models is 0**.
>
> |  | Llama-2-7b | Other LLMs |
> | -------- | ------ | ------ |
> | Llama-2-7b | 18         | 0          |
> | Other LLMs | 0          | 20         |
>
> ---
>
> **Q3:** "It would also be interesting to see what happens when a model is fine-tuned to minimize its REEF score with the model it is derived from."
>
> **A3:** Thank you for your constructive suggestions. **We have already considered the case that fine-tuing a model and minimizing REEF score** (Lines 522-531 and 1241-1280 in the original manuscript). Appendix D indicates that such a fine-tuning strategy fails to evade the detection of REEF.
>
> Beyond the customized loss function in the original paper, **we design two advanced operations and conduct new experiments to evaluate the effectiveness of REEF.**
>
> * We use the customized loss function (Lines 1257-1266 in the original manuscript) to fine-tune different numbers of layers of OPT-1.3B, including fine-tuning only the 12th layer (OPT-1.3B-FT-12th-Layer) and fine-tuning all layers (OPT-1.3B-FT-ALL-Layer) with the E2E NLG Challenge dataset. Then, we calculate their CKA similarity with OPT-1.3b, respectively. According to Figure 3 (Lines 216-232), the CKA similarity between the victim model and unrelated LLMs is usually lower than 0.5. In this way, the low CKA similarity demonstrates that the fine-tuned model has a large representational divergence with OPT-1.3B, thus bypassing the REEF detection. In the following table,  **OPT-1.3B-FT-12th-Layer fails to bypass the detection**; while OPT-1.3B-FT-All-Layer achieves a low CKA similarity, but the high PPL shows that  **the fine-tuned model is unusable**.
>
> |  | CKA Similarity   | PPL              |
> | --------- | --------- | -------- |
> | OPT-1.3B-FT-12th-Layer | **0.8261** | 36.18            |
> | OPT-1.3B-FT-All-Layer  | 0.3270           | **298263** |
>
> * We designed **an alternative approach using Wasserstein loss** to maximize the divergence between the logits of the original and fine-tuned models. This loss function replaced the original customized loss function, with all other settings unchanged (Lines 1257-1267 in the original manuscript). When evaluated on the lambada_openai dataset, the perplexity of the fine-tuned model reached 2375135, rendering the fine-tuned model unusable.
>
> $\mathcal{L} _ {\text{W}} = \max \left( \mathbb{E} _ {x \sim \mathcal{D}} \left[W(\text{LG} _ {\text{ori}}(x), \text{LG} _ {\text{ft}}(x)) \right]\right)$
>
> where$W(\cdot, \cdot)$ represents the Wasserstein distance between two distributions (e.g., logits of the original and fine-tuned models)
>
>
> While our attempts to bypass REEF detection were unsuccessful, exploring additional methods to instantiate the informed adversary remains a promising direction for further validating and ensuring REEF's reliability.

---

> > ### Comment · Reviewer_i9nF · 2024-11-26
> >
> > Thank you for your answers and additional experiments.

---

> > > ### Author Response · Authors · 2024-11-29
> > > **Thanks for your reply!**
> > >
> > > We thank the reviewer for all the valuable comments and questions. We are grateful for your engagement in the rebuttal process.
> > >
> > > Please feel free to share any other insightful suggestions. We would like to improve the paper with any valuable comments.

---

### Official Review · Reviewer_4mRo · 2024-11-04

**Soundness:** 4
**Presentation:** 4
**Contribution:** 4
**Rating:** 10
**Confidence:** 4

**Summary:**

This paper aims to protect LLM's IP by identifying the relationship between the suspect and victim models from the perspective of LLMs’ feature representations. This paper proposes REEF that identifies whether a suspect model is derived from a victim model, given the representations of these two models on certain examples.
The similarity between representations is quantified using Centered Kernel Alignment.
Empirical evaluations of this paper consider:
- Victim model: Llama-2-7b
- Suspect model: quantization and fine-tuned variants of Llama-2-7b as well as unrelated models

**Strengths:**

1. The proposed approach REEF
 - Is training-free, simple and efficient
 - Does not impair the model’s general capabilities
 - Is robust to sequential fine-tuning, pruning, model merging, and permutations
 - Is intuitive as feature representations of fine-tuned victim models are similar to feature representations of the original victim model, while the feature representations of unrelated models exhibit distinct distributions

2. Experimental evaluation
 - considers an informed adversary who is aware of the proposed approach REEF and tries to evade the detection. This paper creates that adversary through the designing of a customized loss function that maximises the representational divergence between models. Results in Appendix D indicate that such fine-tuning seriously damages the model’s general capabilities and renders the fine-tuned models unusable.
- considers comparison with baselines: Weight-based Fingerprinting Methods and Representation-based Fingerprinting Methods
- considers robustness to fine-tuning, pruning, merging, permutation and scaling transformation

**Weaknesses:**

I enjoyed reading this paper, and I am happy with the current version as it has already considered all important studies in its theoretical and empirical analyses. Below, are a few suggestions which could improve the paper further:
1. This paper creates that adversary by designing a customized loss function that maximises the representational divergence between models. Is there any more effective way to create such an informed adversary? For example, designing a better loss function or performing other operations?
2. This paper studies the impact of sample numbers on the performance of REEF. However, there is no evaluation of the type of data points for a given number of samples. Can the performance of REEF be improved if we select/generate samples in a specific way instead of random sample selection?
3. This paper considers one single family of LLMs as a victim model. Generalization of results to other families of victim models would be interesting.
4. One of the observations of this paper is that "CKA from a single layer is sufficient for fingerprint identification." In order to relax the assumption of requiring access to intermediate representations of models for the proposed approach, how much does the detection performance degrade if we use only the last layer?

**Questions:**

See my above questions regarding instantiating the informed adversary, the type of selected examples, generalisation of your results to other families of LLMs as victims, and relaxing the requirements of having access to intermediate representations for detection.

---

> ### Author Response · Authors · 2024-11-24
> **Response to Reviewer 4mRo (Part Ⅰ)**
>
> Thank you for your great efforts in reviewing this paper and for your recognition of our work. We will try our best to answer all your questions. Please let us know if you still have further concerns, or if you are not satisfied with the current responses, so that we can further update the response ASAP.
>
> ---
>
> **Q1:**"This paper creates that adversary by designing a customized loss function that maximizes the representational divergence between models. Is there any more effective way to create such an informed adversary? For example, designing a better loss function or performing other operations?"
>
> **A1:** Thank you for your constructive comments. Beyond the customized loss function proposed in the original manuscript to evade detection by REEF (Lines 522-531 and 1241-1280 in the original manuscript), **we have followed your suggestions to conduct new experiments** to instantiate the informed adversary:
>
> * We use the customized loss function (Lines 1257-1266 in the original manuscript) to fine-tune different numbers of layers of OPT-1.3B, including fine-tuning only the 12th layer (OPT-1.3B-FT-12th-Layer) and fine-tuning all layers (OPT-1.3B-FT-ALL-Layer) with the E2E NLG Challenge dataset. Then, we calculate their CKA similarity with OPT-1.3b, respectively. According to Figure 3 (Lines 216-232), the CKA similarity between the victim model and unrelated LLMs is usually lower than 0.5. In this way, the low CKA similarity demonstrates that the fine-tuned model has a large representational divergence with OPT-1.3B, thus bypassing the REEF detection. In the following table,  **OPT-1.3B-FT-12th-Layer fails to bypass the detection**; while OPT-1.3B-FT-All-Layer achieves a low CKA similarity, but the high PPL shows that  **the fine-tuned model is unusable**.
>
> |   | CKA Similarity   | PPL              |
> | ------------------------ | ------------------ | ------------------ |
> | OPT-1.3B-FT-12th-Layer | **0.8261** | 36.18            |
> | OPT-1.3B-FT-All-Layer  | 0.3270           | **298263** |
>
> * We designed **an alternative approach using Wasserstein loss** to maximize the divergence between the logits of the original and fine-tuned models. This loss function replaced the original customized loss function, with all other settings unchanged (Lines 1257-1267 in the original manuscript ). When evaluated on the lambada_openai dataset, the perplexity of the fine-tuned model reached 2375135, rendering the fine-tuned model unusable.
>
> $\mathcal{L} _ {\text{W}} = \max ( \mathbb{E} _ {x \sim \mathcal{D}} [W(\text{LG} _ {\text{ori}}(x), \text{LG} _ {\text{ft}}(x)) ])$
>
> where$W(\cdot, \cdot)$  represents the Wasserstein distance between two distributions (e.g., logits of the original and fine-tuned models)
>
>
> While our attempts to bypass REEF detection were unsuccessful, exploring additional methods to instantiate the informed adversary remains a promising direction for further validating and ensuring REEF's reliability.
>
> The above experiments and discussions are added in Appendix E.
>
> ---
>
>
> **Q2:**  " Can the performance of REEF be improved if we select/generate samples in a specific way instead of random sample selection?"
>
> **A2:** Thank you for your suggestions. The performance of REEF can be improved by selecting or generating samples in a specific way. As illustrated in Figure 5 (Lines 432-443), REEF shows varying CKA similarities across different datasets, including  *TruthfulQA, SST2, ConfAIde, PKUSafeRLHF, and ToxiGen*. Specifically, **"the gap in the similarity between derived LLMs and unrelated LLMs varies by dataset, e.g., the gap is approximately 0.8 on TruthfulQA and about 0.5 on ToxiGen"** (Lines 506-508). Therefore, selecting samples in specific datasets (e.g.,  **TruthfulQA** ) can improve REEF's performance.
>
> ---

---

> ### Author Response · Authors · 2024-11-24
> **Response to Reviewer 4mRo (Part Ⅱ)**
>
> **Q3:** "This paper considers one single family of LLMs as a victim model. Generalization of results to other families of victim models would be interesting."
>
> **A3:** Thank you for your insightful suggestions. We mainly conduct experiments on Llama, because it is one of the most popular LLMs with extensive subsequent developments. Nevertheless, we have **followed your suggestions to conduct new experiments** on two other families, where **Qwen-2.5-7b** and **Mistral-7b**areadditional victim models. Experimental results in the following table illustrate REEF's robust performance across various scenarios including fine-tuning, pruning, merging, and parameter perturbation on these distinct model families. **This robustness across different families of LLMs highlights REEF's general effectiveness.**
>
> * **Qwen family** : Qwen-2.5-7b as the victim model.
>
> | | Qwen-2.5-7b-coder | Qwen-2.5-7b-pruning | Qwent-7b         | Qwen-2.5-7b-permutation |
> | ---------------- | ------------------- | --------------------- | ------------------ | ------------------------- |
> | PCS            | 0.6769            | 0.0000              | 0.9499           | 0.0000                  |
> | ICS            | 0.9461            | 0.7638              | 0.9989           | 0.9197                  |
> | Logits         | 0.0670            | 0.9999              | 0.8167           | 0.0000                  |
> | **REEF** | **0.9411**  | **0.9785**    | **0.9599** | **1.0000**        |
>
> * **Mistral family** : Mistral-7b as the victim model.
>
> | | Mathstral-7B     | Mistral-7b-pruning | Evollm-jp-7b     | Mistral-7b-permutation |
> | ---------------- | ------------------ | -------------------- | ------------------ | ------------------------ |
> | PCS            | 0.9803           | 0.0000             | 0.9989           | 0.0000                 |
> | ICS            | 0.9883           | 0.6392             | 0.9928           | 0.9847                 |
> | Logits         | 0.3867           | 0.9999             | 0.9999           | 0.0000                 |
> | **REEF** | **0.9344** | **0.9868**   | **0.9516** | **1.0000**       |
>
> The above experiments and discussions are added in Appendix D.
>
> ---
>
> **Q4:** "One of the observations of this paper is that 'CKA from a single layer is sufficient for fingerprint identification.' In order to relax the assumption of requiring access to intermediate representations of models for the proposed approach, how much does the detection performance degrade if we use only the last layer?"
>
> **A4:** Thanks for your comments. We have followed your suggestions to **report the CKA similarity between different layers** of the victim model ( *Llama-2-7b* ) and its fine-tuned models ( *Chinesellama-2-7b and Codellama-2-7b* ), as well as unrelated models ( *Mistral-7b and Qwen-1.5-7b* ) in the following table. **Although  the CKA similarity  at the last layer shows a decrease  compared to mid-layers, the similarity between the victim model and its derived LLMs remains higher than with unrelated LLMs** , indicating that REEF is still effective. However, we recommend using representations from intermediate layers for detection, as they provide superior discriminative power compared to the last layer.
>
> | | 8th Layer | 16th Layer | 24th Layer | **32nd Layer** |
> | ------------------- | ----------- | ------------ | ------------ | ---------------------- |
> | Chinesellama-2-7b | 0.9992    | 0.9978     | 0.9947     | **0.9330**     |
> | CodeLlama-2-7b    | 0.9965    | 0.9960     | 0.9917     | **0.9122**     |
> | Mistral-7b        | 0.1548    | 0.1978     | 0.3066     | **0.6321**     |
> | Qwen-1.5-7b       | 0.1626    | 0.2061     | 0.2840     | **0.6189**     |

---

> > ### Comment · Reviewer_4mRo · 2024-11-25
> > **Thank you!**
> >
> > Thanks for the detailed response, and for performing new experiments which were optional!

---

> > > ### Author Response · Authors · 2024-11-26
> > > **Thanks for your reply!**
> > >
> > > We are really grateful for your constructive comments and the reply. The comments indeed help us further improve the paper.

---

### Meta-Review · Area_Chair_oCSk · 2024-12-22

**Metareview:**

This paper presents a new method for fingerprinting LLMs called REEF. The method leverages feature representations computed by the suspect and victim LLMs, and computes kernel alignment similarity to determine if the suspect model is derived from the victim model. Some advantages of REEF that reviewers cited include training-freeness, robustness to modifications such as fine-tuning and model pruning, and intuitiveness of the method. While reviewers also raised some weaknesses, overall evaluation of the paper is overwhelmingly positive, and AC is happy to recommend acceptance.

**Additional Comments On Reviewer Discussion:**

Reviewers and authors mostly discussed minor weaknesses. Reviewer i9nF raised a question regarding the method's false positive rate, which the authors addressed by providing a clarification statement and additional experiment results.

---

### Decision · Program_Chairs · 2025-01-22

Accept (Oral)